# Center Smoothing: Certified Robustness for Networks with Structured Outputs

**Aounon Kumar**
University of Maryland
aounon@umd.edu

**Tom Goldstein**
University of Maryland
tomg@cs.umd.edu

## Abstract

The study of provable adversarial robustness has mostly been limited to classification tasks and models with one-dimensional real-valued outputs. We extend the scope of certifiable robustness to problems with more general and structured outputs like sets, images, language, etc. We model the output space as a metric space under a distance/similarity function, such as intersection-over-union, perceptual similarity, total variation distance, etc. Such models are used in many machine learning problems like image segmentation, object detection, generative models, image/audio-to-text systems, etc. Based on a robustness technique called randomized smoothing, our *center smoothing* procedure can produce models with the guarantee that the change in the output, as measured by the distance metric, remains small for any norm-bounded adversarial perturbation of the input. We apply our method to create certifiably robust models with disparate output spaces – from sets to images – and show that it yields meaningful certificates without significantly degrading the performance of the base model.

## 1   Introduction

The study of adversarial robustness in machine learning (ML) has gained a lot of attention ever since deep neural networks (DNNs) have been demonstrated to be vulnerable to adversarial attacks. These attacks are generated by making tiny perturbations of the input that can completely alter a model's predictions [56, 46, 23, 35]. They can significantly degrade the performance of a model, like an image classifier, and make it output almost any class of the attacker's choice. However, these attacks are not limited just to classification problems. They have also been shown to exist for DNNs with structured outputs like text, images, probability distributions, sets, etc. For instance, automatic speech recognition systems can be attacked with 100% success rate to output any phrase of the attackers choice [10]. Similar attacks can cause neural image captioning systems to produce specific target captions with high success-rate [11]. Quality of image segmentation models have been shown to degrade severely under adversarial attacks [2, 27, 30]. Facial recognition systems can be deceived to evade detection, impersonate authorized individuals and even render them completely ineffective [59, 55, 20]. Image reconstruction models have been targeted to introduce unwanted artefacts or miss important details, such as tumors in MRI scans, through adversarial inputs [1, 50, 8, 12]. Super-resolution systems can be made to generate distorted images that can in turn deteriorate the performance of subsequent tasks that rely on the high-resolution outputs [14, 63]. Deep neural network based policies in reinforcement learning problems also have been shown to succumb to imperceptible perturbations in the state observations [21, 29, 4, 48]. Such widespread presence of adversarial attacks is concerning as it threatens the use of deep neural networks in critical systems, such as facial recognition, self-driving vehicles, medical diagnosis, etc., where safety, security and reliability are of utmost importance.

35th Conference on Neural Information Processing Systems (NeurIPS 2021).

Adversarial defenses have mostly focused on classification tasks [34, 6, 26, 17, 44, 25, 22]. Certified defenses based on convex-relaxation [61, 49, 53, 13, 54], interval-bound propagation [24, 28, 18, 47] and randomized smoothing [15, 36, 42, 51] that guarantee that the predicted class will remain the same in a certified region around the input point have also been studied. Compared to empirical robustness methods that are often shown to be broken by stronger attacks [9, 3, 58], procedures with provable robustness guarantees are of special importance to the study of robustness in ML as their guarantees hold regardless of improvements in attack strategies. Among these approaches, certified defenses based on randomized smoothing have been show to scale up to high-dimensional inputs, such as images, and does not need to make assumptions about the underlying model. The robustness certificates produced by these defenses are probabilistic, meaning that they hold with high probability and not absolute certainty.

Unlike classification problems, where certificates guarantee that the predicted class remains unchanged under bounded-size perturbations, it is not immediately obvious what the goal of robustness should be for problems with structured outputs like images, text, sets, etc. While accuracy is the standard quality measure for classification, more complex tasks may require other quality metrics like total variation for images, intersection over union for object localization, earth-mover distance for distributions, etc. In general, neural networks can be cast as functions of the type $f : \mathbb{R}^k \to (M, d)$ which map a $k$ dimensional real-valued space into a metric space $M$ with distance function $d : M \times M \to \mathbb{R}_{\geq 0}$. In this work, we design a randomized smoothing based technique to obtain provable robustness for functions of this type with minimal assumptions on the distance metric $d$. We generate a robust version $\bar{f}$ such that the change in its output, as measured by $d$, is small for a small change in its input. More formally, given an input $x$ and an $\ell_2$-perturbation size $\epsilon_1$, we produce a value $\epsilon_2$ with the guarantee that, with high probability,

$$\forall x' \text{ s.t. } \|x - x'\|_2 \leq \epsilon_1, \ d(\bar{f}(x), \bar{f}(x')) \leq \epsilon_2.$$

**Our contributions:** We develop *center smoothing*, a procedure to make functions like $f$ provably robust against adversarial attacks. For a given input $x$, center smoothing samples a collection of points in the neighborhood of $x$ using a Gaussian smoothing distribution, computes the function $f$ on each of these points and returns the center of the smallest ball enclosing at least half the points in the output space (see figure 1). Computing the minimum enclosing ball in the output space is equivalent to solving the 1-center problem with outliers (hence the name of our procedure), which is an NP-complete problem for a general metric [52]. We approximate it by computing the point that has the smallest median distance to all the other points in the sample. We show that the output of the smoothed function is robust to input perturbations of bounded $\ell_2$-size. We restrict the input perturbations to be inside an $\ell_2$-ball as the main focus

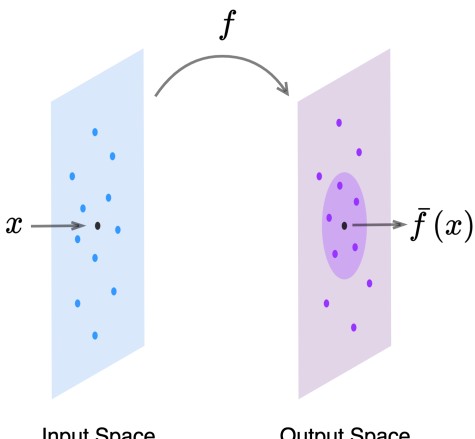

Figure 1: Center smoothing.

of this work is on the output space of $f$. However, our method does not critically rely on the $\ell_2$ threat model or Gaussian smoothing noise, and can be adapted to other perturbations types and smoothing distributions. Although we define the output space as a metric, our proofs only require the symmetry property and triangle inequality to hold. Thus, center smoothing can also be applied to pseudometric distances that need not satisfy the identity of indiscernibles. Many distances defined for images, such as total variation, cosine distance, perceptual distances, etc., fall under this category. Center smoothing steps outside the world of $\ell_p$ metrics, and certifies robustness in metrics like IoU/Jaccard distance for object localization, and total-variation, which is a good measure of perceptual similarity for images. In our experiments, we show that this method can produce meaningful certificates for a wide variety of output metrics without significantly compromising the quality of the base model.

**Related Work:** Randomized smoothing has been extensively used for provable adversarial robustness in the classification setting to defend against different $\ell_p$ [15, 36, 51, 57, 43, 41, 37, 40] and non-$\ell_p$ [38, 39] threat models. Beyond classification, it has also been used for certifying the median output of regression models [62] and the expected softmax scores of neural networks [33]. Smoothing a bounded vector-valued function by taking the mean of the output vectors has been shown to have a

bounded Lipschitz constant when both input and output spaces are $\ell_2$-metrics [60]. Center smoothing does not require the base function to be bounded because the minimum enclosing ball is resistant to outliers. Moving an outlier point away from this ball does not affect the output of the smoothed function. On the other hand, smoothing techniques that compute the mean of the output samples are more susceptible to outliers as changing any of the samples can alter the mean. Recently, a provable defense for segmentation tasks was developed by certifying each individual pixel of the output using randomized smoothing [19]. Due to the accumulating uncertainty over individual certifications, it is difficult to produce guarantees for large images, often leading to certified outputs with ambiguous pixels. Center smoothing bypasses this challenge by directly certifying the similarity between a clean segmentation output and an adversarial one under a metric such as intersection over union.

## 2 Preliminaries and Notations

Given a function $f : \mathbb{R}^k \to (M, d)$ and a distribution $\mathcal{D}$ over the input space $\mathbb{R}^k$, let $f(\mathcal{D})$ denote the probability distribution of the output of $f$ in $M$ when the input is drawn from $\mathcal{D}$. For a point $x \in \mathbb{R}^k$, let $x + \mathcal{P}$ denote the probability distribution of the points $x + \delta$ where $\delta$ is a smoothing noise drawn from a distribution $\mathcal{P}$ over $\mathbb{R}^k$ and let $X$ be the random variable for $x + \mathcal{P}$. For elements in $M$, define $\mathcal{B}(z, r) = \{z' \mid d(z, z') \leq r\}$ as a ball of radius $r$ centered at $z$. Define a smoothed version of $f$ under $\mathcal{P}$ as the center of the ball with the smallest radius in $M$ that encloses at least half of the probability mass of $f(x + \mathcal{P})$, i.e.,

$$\bar{f}_{\mathcal{P}}(x) = \operatorname*{argmin}_{z} r \text{ s.t. } \mathbb{P}[f(X) \in \mathcal{B}(z, r)] \geq \frac{1}{2}.$$

If there are multiple balls with the smallest radius satisfying the above condition, return one of the centers arbitrarily. Let $r_{\mathcal{P}}^*(x)$ be the value of the minimum radius. Hereafter, we ignore the subscripts and superscripts in the above definitions whenever they are obvious from context. In this work, we sample the noise vector $\delta$ from an i.i.d Gaussian distribution of variance $\sigma^2$ in each dimension, i.e., $\delta \sim \mathcal{N}(0, \sigma^2 I)$.

### 2.1 Gaussian Smoothing

Cohen et al. in 2019 showed that a classifier $h : \mathbb{R}^k \to \mathcal{Y}$ smoothed with a Gaussian noise $\mathcal{N}(0, \sigma^2 I)$ as,

$$\bar{h}(x) = \operatorname*{argmax}_{c \in \mathcal{Y}} \mathbb{P}\left[h(x + \delta) = c\right],$$

where $\mathcal{Y}$ is a set of classes, is certifiably robust to small perturbations in the input. Their certificate relied on the fact that, if the probability of sampling from the top class at $x$ under the smoothing distribution is $p$, then for an $\ell_2$ perturbation of size at most $\epsilon$, the probability of the top class is guaranteed to be at least

$$p_\epsilon = \Phi(\Phi^{-1}(p) - \epsilon/\sigma), \tag{1}$$

where $\Phi$ is the CDF of the standard normal distribution $\mathcal{N}(0, 1)$. This bound applies to any $\{0, 1\}$-function over the input space $\mathbb{R}^k$, i.e., if $\mathbb{P}[h(x) = 1] = p$, then for any $\epsilon$-size perturbation $x'$, $\mathbb{P}[h(x') = 1] \geq p_\epsilon$.

We use this bound to generate robustness certificates for center smoothing. We identify a ball $\mathcal{B}(\bar{f}(x), R)$ of radius $R$ enclosing a very high probability mass of the output distribution. One can define a function that outputs one if $f$ maps a point to inside $\mathcal{B}(\bar{f}(x), R)$ and zero otherwise. The bound in (1) gives us a region in the input space such that for any point inside it, at least half of the mass of the output distribution is enclosed in $\mathcal{B}(\bar{f}(x), R)$. We show in section 3 that the output of the smoothed function for a perturbed input is guaranteed to be within a constant factor of $R$ from the output of the original input.

## 3 Center Smoothing

As defined in section 2, the output of $\bar{f}$ is the center of the smallest ball in the output space that encloses at least half the probability mass of the $f(x + \mathcal{P})$. Thus, in order to significantly change the output, an adversary has to find a perturbation such that a majority of the neighboring points map far away from $\bar{f}(x)$. However, for a function that is roughly accurate on most points around $x$, a

small perturbation in the input cannot change the output of the smoothed function by much, thereby making it robust.

For an $\ell_2$ perturbation size of $\epsilon_1$ of an input point $x$, let $R$ be the radius of a ball around $\bar{f}(x)$ that encloses more than half the probability mass of $f(x' + \mathcal{P})$ for all $x'$ satisfying $\|x - x'\|_2 \leq \epsilon_1$, i.e.,

$$\forall x' \text{ s.t. } \|x - x'\|_2 \leq \epsilon_1, \ \mathbb{P}[f(X') \in \mathcal{B}(\bar{f}(x), R)] > \frac{1}{2}, \tag{2}$$

where $X' \sim x' + \mathcal{P}$. Basically, $R$ is the radius of a ball around $\bar{f}(x)$ that contains at least half the probability mass of $f(x' + \mathcal{P})$ for any $\epsilon_1$-size perturbation $x'$ of $x$. Then, we have the following robustness guarantee on $\bar{f}$:

**Theorem 1.** *For all $x'$ such that $\|x - x'\|_2 \leq \epsilon_1$,*

$$d(\bar{f}(x), \bar{f}(x')) \leq 2R.$$

*Proof.* Consider the balls $\mathcal{B}(\bar{f}(x'), r^*(x'))$ and $\mathcal{B}(\bar{f}(x), R)$ (see figure 2). From the definition of $r^*(x')$ and $R$, we know that the sum of the probability masses of $f(x' + \mathcal{P})$ enclosed by the two balls must be strictly greater than one. Thus, they must have an element $y$ in common. Since $d$ satisfies the triangle inequality, we have:

$$d(\bar{f}(x), \bar{f}(x')) \leq d(\bar{f}(x), y) + d(y, \bar{f}(x'))$$
$$\leq R + r^*(x').$$

Since, the ball $\mathcal{B}(\bar{f}(x), R)$ encloses more than half of the probability mass of $f(x + \mathcal{P})$, the minimum ball with at least half the probability mass cannot have a radius greater than $R$, i.e., $r^*(x') \leq R$. Therefore, $d(\bar{f}(x), \bar{f}(x')) \leq 2R$. $\qquad\square$

The above result, in theory, gives us a smoothed version of $f$ with a provable guarantee of robustness. However, in practice, it may not be feasible to obtain $\bar{f}$ just from samples of $f(x + \mathcal{P})$. Instead, we will use some procedure that approximates the smoothed output with high probability. For some $\Delta \in [0, 1/2]$, let $\hat{r}(x, \Delta)$ be the radius of the smallest ball that encloses at least $1/2 + \Delta$ probability mass of $f(x + \mathcal{P})$, i.e.,

$$\hat{r}(x, \Delta) = \min_{z'} r \text{ s.t. } \mathbb{P}[f(X) \in \mathcal{B}(z', r)] \geq \frac{1}{2} + \Delta.$$

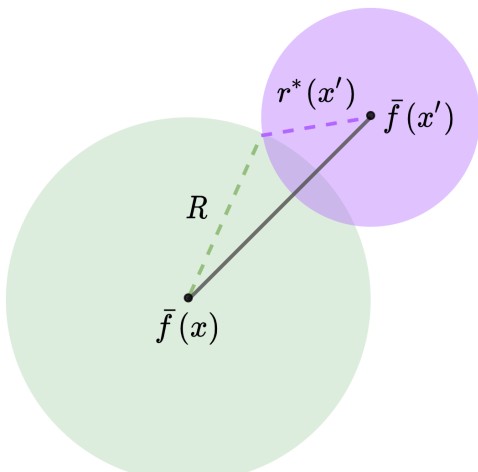

Now define a probabilistic approximation $\hat{f}(x)$ of the smoothed function $\bar{f}$ to be a point $z \in M$, which with probability at least $1 - \alpha_1$ (for $\alpha_1 \in [0, 1]$), encloses at least $1/2 - \Delta$ probability mass of $f(x + \mathcal{P})$ within a ball of radius $\hat{r}(x, \Delta)$. Formally, $\hat{f}(x)$ is a point $z \in M$, such that, with at least $1 - \alpha_1$ probability,

$$\mathbb{P}\left[f(X) \in \mathcal{B}(z, \hat{r}(x, \Delta))\right] \geq \frac{1}{2} - \Delta.$$

Figure 2: Robustness guarantee.

Defining $\hat{R}$ to be the radius of a ball centered at $\hat{f}(x)$ that satisfies:

$$\forall x' \text{ s.t. } \|x - x'\|_2 \leq \epsilon_1, \ \mathbb{P}[f(X') \in \mathcal{B}(\hat{f}(x), \hat{R})] > \frac{1}{2} + \Delta, \tag{3}$$

we can write a probabilistic version of theorem 1,

**Theorem 2.** *With probability at least $1 - \alpha_1$,*

$$\forall x' \text{ s.t. } \|x - x'\|_2 \leq \epsilon_1, \ d(\hat{f}(x), \hat{f}(x')) \leq 2\hat{R},$$

The proof of this theorem is in the appendix, and logically parallels the proof of theorem 1.

| **Algorithm 1** Smooth | **Algorithm 2** Certify |
|---|---|
| **Input:** $x \in \mathbb{R}^k, \sigma, \Delta, \alpha_1$. | **Input:** $x \in \mathbb{R}^k, \epsilon_1, \sigma, \Delta, \alpha_1, \alpha_2$. |
| **Output:** $z \in M$. | **Output:** $\epsilon_2 \in \mathbb{R}$. |
| Set $Z = \{z_i\}_{i=1}^n$ s.t. $z_i \sim f(x + \mathcal{N}(0, \sigma^2 I))$. | Compute $\hat{f}(x)$ using algorithm 1. |
| Set $\Delta_1 = \sqrt{\ln(2/\alpha_1)/2n}$. | Set $Z = \{z_i\}_{i=1}^m$ s.t. $z_i \sim f(x + \mathcal{N}(0, \sigma^2 I))$. |
| Compute $z = \beta\text{-MEB}(Z, 1/2)$. | Compute $\tilde{\mathcal{R}} = \{d(\hat{f}(x), f(z_i)) \mid z_i \in Z\}$. |
| Re-sample $Z$. | Set $p = \Phi(\Phi^{-1}(1/2 + \Delta) + \epsilon_1/\sigma)$. |
| Compute $p_{\Delta_1}$. | Set $q = p + \sqrt{\ln(1/\alpha_2)/2m}$. |
| Set $\Delta_2 = 1/2 - p_{\Delta_1}$. | Set $\hat{R} = q$th-quantile of $\tilde{\mathcal{R}}$. |
| If $\Delta < \max(\Delta_1, \Delta_2)$, discard $z$ and abstain. | Set $\epsilon_2 = (1 + \beta)\hat{R}$. |

### 3.1 Computing $\hat{f}$

For an input $x$ and a given value of $\Delta$, sample $n$ points independently from a Gaussian distribution $x + \mathcal{N}(0, \sigma^2 I)$ around the point $x$ and compute the function $f$ on each of these points. Let $Z = \{z_1, z_2, \ldots, z_n\}$ be the set of $n$ samples of $f(x + \mathcal{N}(0, \sigma^2 I))$ produced in the output space. Compute the minimum enclosing ball $\mathcal{B}(z, r)$ that contains at least half of the points in $Z$. The following lemma bounds the radius $r$ of this ball by the radius of the smallest ball enclosing at least $1/2 + \Delta_1$ probability mass of the output distribution (proof in appendix).

**Lemma 1.** *With probability at least* $1 - e^{-2n\Delta_1^2}$,

$$r \leq \hat{r}(x, \Delta_1).$$

Now, sample a fresh batch of $n$ random points. Let $p_{\Delta_1} = \rho - \Delta_1$, where $\rho$ is the fraction of points that fall inside $\mathcal{B}(z, r)$. Then, by Hoeffding's inequality, with probability at least $1 - e^{-2n\Delta_1^2}$,

$$\mathbb{P}[f(X) \in \mathcal{B}(z, r)] \geq p_{\Delta_1}.$$

Let $\Delta_2 = 1/2 - p_{\Delta_1}$. If $\max(\Delta_1, \Delta_2) \leq \Delta$, the point $z$ satisfies the conditions in the definition of $\hat{f}$, with at least $1 - 2e^{-2n\Delta_1^2}$ probability. If $\max(\Delta_1, \Delta_2) > \Delta$, discard the computed center $z$ and abstain. In our experiments, we select $\Delta_1, n$ and $\alpha_1$ appropriately so that the above process succeeds easily.

Computing the minimum enclosing ball $\mathcal{B}(z, r)$ exactly can be computationally challenging, as for certain metrics, it is known to be NP-complete [52]. Instead, we approximate it by computing a ball $\beta\text{-MEB}(Z, 1/2)$ that contains at least half the points in $Z$, but has a radius that is within a $\beta$ factor of the optimal radius $r$. We modify theorem 1 to account for this approximation (see appendix for proof).

**Theorem 3.** *With probability at least* $1 - \alpha_1$,

$$\forall x' \text{ s.t. } \|x - x'\|_2 \leq \epsilon_1, \ d(\hat{f}(x), \hat{f}(x')) \leq (1 + \beta)\hat{R}$$

*where* $\alpha_1 = 2e^{-2n\Delta_1^2}$.

We use a simple approximation that works for all metrics and achieves an approximation factor of two, producing a certified radius of $3\hat{R}$. It computes a point from the set $Z$, instead of a general point in $M$, that has the minimum median distance from all the points in the set (including itself). This can be achieved using $O(n^2)$ pair-wise distance computations. To see how the factor 2-approximation is achieved, consider the optimal ball with radius $r$. By triangle inequality of $d$, each pair of points is at most $2r$ distance from each other. Thus, a ball with radius $2r$, centered at any one of these points will cover every other point in the optimal ball. Better approximations can be obtained for specific norms, e.g., there exists a $(1 + \epsilon)$-approximation algorithm for the $\ell_2$ norm [7]. For graph distances or when the support of the output distribution is a small discrete set of points, the optimal radius can be computed exactly using the above algorithm. The smoothing procedure is outlined in algorithm 1.

### 3.2 Certifying $\hat{f}$

Given an input $x$, compute $\hat{f}(x)$ as described above. Now, we need to compute a radius $\hat{R}$ that satisfies condition 3. As per bound 1, in order to maintain a probability mass of at least $1/2 + \Delta$ for

any $\epsilon_1$-size perturbation of $x$, the ball $\mathcal{B}(\hat{f}(x), \hat{R})$ must enclose at least

$$p = \Phi\left(\Phi^{-1}\left(\frac{1}{2} + \Delta\right) + \frac{\epsilon_1}{\sigma}\right) \tag{4}$$

probability mass of $f(x + \mathcal{P})$. Again, just as in the case of estimating $\bar{f}$, we may only compute $\hat{R}$ from a finite number of samples $m$ of the distribution $f(x + \mathcal{P})$. For each sample $z_i \sim x + \mathcal{P}$, we compute the distance $d(\hat{f}(x), f(z_i))$ and set $\hat{R}$ to be the $q$th-quantile $\tilde{R}_q$ of these distances for a $q$ that is slightly greater than $p$ (see equation 5 below). The $q$th-quantile $\tilde{R}_q$ is a value larger than at least $q$ fraction of the samples. We set $q$ as,

$$q = p + \sqrt{\frac{\ln(1/\alpha_2)}{2m}}, \tag{5}$$

for some small $\alpha_2 \in [0, 1]$. This guarantees that, with high probability, the ball $\mathcal{B}(\hat{f}(x), \tilde{R}_q)$ encloses at least $p$ fraction of the probability mass of $f(x + \mathcal{P})$. We prove the following lemma by bounding the cumulative distribution function of the distances of $f(z_i)$s from $\hat{f}(x)$ using the Dvoretzky–Kiefer–Wolfowitz inequality.

**Lemma 2.** *With probability* $1 - \alpha_2$,

$$\mathbb{P}\left[f(X) \in \mathcal{B}(\hat{f}(x), \tilde{R}_q)\right] > p.$$

Combining with theorem 3, we have the final certificate:

$$\forall x' \text{ s.t. } \|x - x'\|_2 \leq \epsilon_1, \ d(\hat{f}(x), \hat{f}(x')) \leq (1 + \beta)\hat{R},$$

with probability at least $1 - \alpha$, for $\alpha = \alpha_1 + \alpha_2$. In our experiments, we set $\alpha_1 = \alpha_2 = 0.005$ to achieve an overall success probability of $1 - \alpha = 0.99$, and calculate the required $\Delta_1, \Delta_2$ and $q$ values accordingly. We set $\Delta$ to be as small as possible without violating $\max(\Delta_1, \Delta_2) \leq \Delta$ too often. We use a $\beta = 2$-approximation for computing the minimum enclosing ball in the smoothing step. Algorithm 2 provides the pseudocode for the certification procedure.

## 4 Relaxing Metric Requirements

Although we defined our procedure for metric outputs, our analysis does not critically use all the properties of a metric. For instance, we do not require $d(z_1, z_2)$ to be strictly greater than zero for $z_1 \neq z_2$. An example of such a distance measure is the total variation distance that returns zero for two vectors that differ by a constant amount on each coordinate. Our proofs do implicitly use the symmetry property, but asymmetric distances can be converted to symmetric ones by taking the sum or the max of the distances in either directions. Perhaps the most important property of metrics that we use is the triangle inequality as it is critical for the robustness guarantee of the smoothed function. However, even this constraint may be partially relaxed. It is sufficient for the distance function $d$ to satisfy the triangle inequality approximately, i.e., $d(a, c) \leq \gamma(d(a, b) + d(b, c))$, for some constant $\gamma$. The theorems and lemmas can be adjusted to account for this approximation, e.g., the bound in theorem 1 will become $2\gamma R$. A commonly used distance measure for comparing images and documents is the cosine distance defined as the inner-product of two vectors after normalization. This distance can be show to be proportional to the squared Euclidean distance between the normalized vectors which satisfies the relaxed version of triangle inequality for $\gamma = 2$.

These relaxations extend the scope of center smoothing to many commonly used distance measures that need not necessarily satisfy all the metric properties. For instance, perceptual distance metrics measure the distance between two images in some feature space rather than image space. Such distances align well with human judgements when the features are extracted from a deep neural network [65] and are considered more natural measures for image similarity. For two images $I_1$ and $I_2$, let $\phi(I_1)$ and $\phi(I_2)$ be their feature representations. Then, for a distance function $d$ in the feature space that satisfies the relaxed triangle inequality, we can define a distance function $d_\phi(I_1, I_2) = d(\phi(I_1), \phi(I_2))$ in the image space, which also satisfies the relaxed triangle inequality. For any image $I_3$,

$$\begin{aligned}
d_\phi(I_1, I_2) &= d(\phi(I_1), \phi(I_2)) \\
&\leq \gamma\left(d(\phi(I_1), \phi(I_3)) + d(\phi(I_3), \phi(I_2))\right) \\
&= \gamma\left(d_\phi(I_1, I_3) + d_\phi(I_3, I_2)\right).
\end{aligned}$$

# 5 Experiments

We apply center smoothing to certify a wide range of output metrics: Jaccard distance based on intersection over union (IoU) of sets, total variation distances for images, and perceptual distance. We certify the bounding box generated by a face detector – a key component of most facial recognition systems – by guaranteeing the minimum overlap (measured using IoU) it must have with the output under an adversarial perturbation of the input. For instance, if $\epsilon_1 = 0.2$, the Jaccard distance (1-IoU) is guaranteed to be bounded by 0.2, which implies that the bounding box of a perturbed image must have at least 80% overlap with that of the clean image. We use a pre-trained face detection model for this experiment. We certify the perceptual distance of the output of a generative model (trained on ImageNet) that produces $128 \times 128$ RGB images using a high-dimensional version of the smoothing procedure Smooth-HD described in the appendix. For total variation distance, we use simple, easy-to-train convolutional neural network based dimensionality reduction (autoencoder) and image reconstruction models. Our goal is to demonstrate the effectiveness of our method for a wide range of applications and so, we place less emphasis on the performance of the underlying models being smoothed. In each case, we show that our method is capable of generating certified guarantees without significantly degrading the performance of the underlying model. We provide additional experiments for other metrics and parameter settings in the appendix.

As is common in the randomized smoothing literature, we train our base models (except for the pre-trained ones) on noisy data with different noise levels $\sigma_{train} = 0.1, 0.2, \ldots, 0.5$ to make them more robust to input perturbations. We keep the smoothing noise $\sigma$ of the robust model same as the training noise $\sigma_{train}$ of the base model. We use $n = 10^4$ samples to estimate the smoothed function and $m = 10^6$ samples to generate certificates, unless stated otherwise. We set $\Delta = 0.05, \alpha_1 = 0.005$ and $\alpha_2 = 0.005$ as discussed in previous sections. We grow the smoothing noise $\sigma$ linearly with the input perturbation $\epsilon_1$. Specifically, we maintain $\epsilon_1 = h\sigma$ for different values of $h = 2, 1$ and $1.5$ in our experiments. We plot the median certified output radius $\epsilon_2$ and the median smoothing error, defined as the distance between the outputs of the base model and the smoothed model $d(f(x), \hat{f}(x))$, of fifty random test examples for different values of $\epsilon_1$. In all our experiments, we observe that both these quantities increase as the input radius $\epsilon_1$ increases, but the smoothing error remains significantly below the certified output radius. Also, increasing the value of $h$ improves the quality of the certificates (lower $\epsilon_2$). This could be due to the fact that for a higher $h$, the smoothing noise $\sigma$ is lower (keeping $\epsilon_1$ constant), which means that the radius of the minimum enclosing ball in the output space is smaller leading to a tighter certificate. However, setting $h$ too high can cause the value of $q$ in equation 5 to exceed one ($q$ depends on $p$, which in turn depends on $h$ in eq. 4), leading the certification procedure (algorithm 2) to fail. We ran all our experiments on a single NVIDIA GeForce RTX 2080 Ti GPU in an internal cluster. Each of the fifty examples we certify took somewhere between 1-3 minutes depending on the underlying model.

## 5.1 Jaccard distance

It is known that facial recognition systems can be deceived to evade detection, impersonate authorized individuals and even render completely ineffective [59, 55, 20]. Most facial recognition systems first detect a region that contains a persons face, e.g. a bounding box, and then uses facial features to identify the individual in the image. To evade detection, an attacker may seek to degrade the quality of the bounding boxes produced by the detector and can even cause it to detect no box at all. Bounding boxes are often interpreted as sets and the their quality is measured as the amount of overlap with the desired output. When no box is output, we say the overlap is zero. The overlap between two sets is defined as the ratio of the size of the intersection between them to the size of their union (IoU). Thus, to certify the robustness of the output of a face detector, it makes sense to bound the worst-case IoU of the output of an adversarial input to that of a clean input. The corresponding distance function, known as Jaccard distance, is defined as $1 - IoU$ which defines a metric over the universe of sets.

$$IoU(A, B) = \frac{|A \cap B|}{|A \cup B|}, \quad d_J(A, B) = 1 - IoU(A, B) = 1 - \frac{|A \cap B|}{|A \cup B|}.$$

In this experiment, we certify the output of a pre-trained face detection model MTCNN [64] on the CelebA face dataset [45]. We set $n = 5000$ and $m = 10000$, and use default values for other parameters discussed above. Figure 3a plots the certified output radius $\epsilon_2$ and the smoothing error for $h = \epsilon_1/\sigma = 1$ and $2$ for $\epsilon_1 = 0.1, 0.2, \ldots, 0.5$. Certifying the Jaccard distance allows us to certify

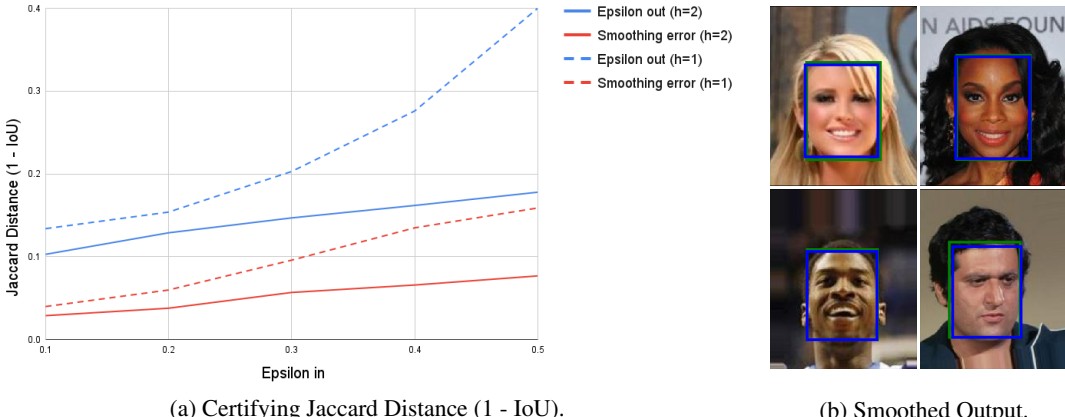

(a) Certifying Jaccard Distance (1 - IoU).

(b) Smoothed Output.

Figure 3: Face Detection on CelebA using MTCNN detector: Part (a) plots the certified output radius $\epsilon_2$ and the smoothing error for $h = 1$ and 2. Part (b) compares the smoothed output (blue box) to the output of the base model (green box, mostly hidden behind the blue box) showing a significant overlap.

IoU as well, e.g., for $h = 2$, $\epsilon_2$ is consistently below 0.2 which means that even the worst bounding box under adversarial perturbation of the input has an overlap of at least 80% with the box for the clean input. The low smoothing error shows that the performance of the base model does not drop significantly as the actual output of the smoothed model has a large overlap with that of the base model. Figure 3b compares the outputs of the smoothed model (blue box) and the base model (green box). For most of the images, the blue box overlaps with the green one almost perfectly.

## 5.2 Perceptual Distance

Deep generative models like GANs and VAEs have been shown to be vulnerable to adversarial attacks [31]. One attack model is to produce an adversarial example that is close to the original input in the latent space, measured using $\ell_2$-norm. The goal is to make the model generate a different looking image using a latent representation that is close to that of the original image. We apply center smoothing to a generative adversarial network BigGAN pre-trained on ImageNet images [5]. We use the version of the GAN that generates $128 \times 128$ resolution ImageNet images from a set of 128 latent variables. Since we are interested in producing similar looking images for similar latent representations, a good output metric would be the perceptual distance between two images measured by LPIPS metric [65]. This distance function takes in two images, passes them through a deep neural network, such as VGG, and computes a weighted sum of the square of the differences of the activations (after some normalization) produced by the two images. The process can be thought of as generating two feature vectors $\phi_1$ and $\phi_2$ for the two input images $I_1$ and $I_2$ respectively, then computing a weighted sum of the element-wise square of the differences between the two feature vectors, i.e.,

$$d(I_1, I_2) = \sum_i w_i (\phi_{1i} - \phi_{2i})^2$$

The square of differences metric can be shown to follow the relaxed triangle inequality for $\gamma = 2$. Therefore, the the final bound on the certified output radius will be $\gamma(1 + 2\gamma)\hat{R} = 10\hat{R}$. Figure 4a plots the median smoothing error and certified output radius $\epsilon_2$ for fifty randomly picked latent vectors for $\epsilon_1 = 0.01, 0.02, \ldots, 0.05$ and $h = 1, 1.5$. For these experiments, we set $n = 2000, m = 10^4$ and $\Delta = 0.8$. We use the modified smoothing procedure Smooth-HD (see appendix) for high-dimensional outputs with a small batch size of 150 to accommodate the samples in memory. It takes about three minutes to smooth and certify each input on a single NVIDIA GeForce RTX 2080 Ti GPU in an internal cluster. Due to the higher factor of ten in the certified output radius in this case compared to our other experiments where the factor is three, the certified output radius increases faster with the input radius $\epsilon_1$, but the smoothing error remains low showing that, in practice, the method does not significantly degrade the performance of the base model. Figure 4b shows that, visually, the smoothed output is not very different from the output of the base model. The input radii we certify for are lower in this case than our other experiments due to the low dimensionality (only 128 dimensions) of the input (latent) space as compared to the input (image) spaces in our other experiments.

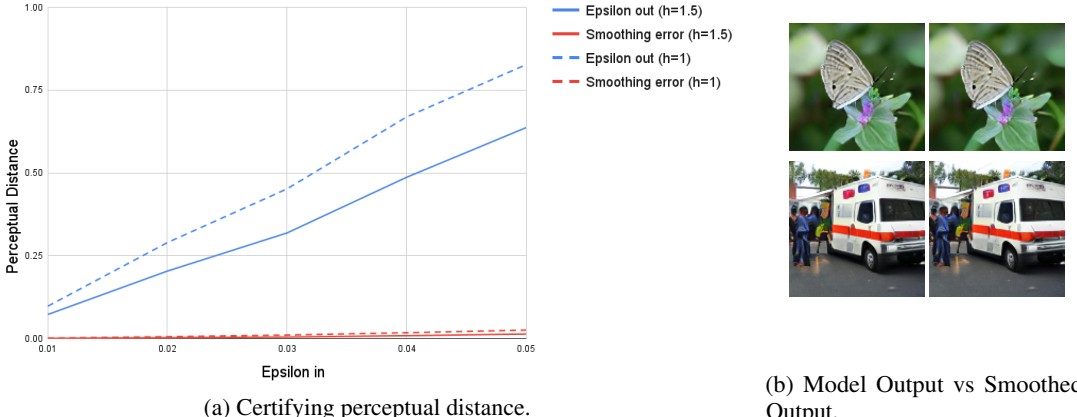

(a) Certifying perceptual distance.

(b) Model Output vs Smoothed Output.

Figure 4: Generative model for ImageNet: Part (a) plots the certified output radius $\epsilon_2$ and the smoothing error for $h = 1$ and 1.5. Part (b) compares the output of the base model to that of the smoothed model.

## 5.3 Total Variation Distance

The total variation norm of a vector $x$ is defined as the sum of the magnitude of the difference between pairs of coordinates defined by a *neighborhood* set $N$. For a 1-dimensional array $x$ with $k$ elements, one can define the neighborhood as the set of consecutive elements.

$$TV(x) = \sum_{(i,j) \in N} |x_i - x_j|, \quad TV_{1D}(x) = \sum_{i=1}^{k-1} |x_i - x_{i+1}|.$$

Similarly, for a grayscale image represented by a $h \times w$ 2-dimensional array $x$, the neighborhood can be defined as the next element (pixel) in the row/column. In case of an RGB image, the difference between the neighboring pixels is a vector, whose magnitude can be computed using an $\ell_p$-norm. For, our experiments we use the $\ell_1$-norm.

$$TV_{RGB}(x) = \sum_{i=1}^{h-1} \sum_{j=1}^{w-1} \|x_{i,j} - x_{i+1,j}\|_1 + \|x_{i,j} - x_{i,j+1}\|_1$$

The total variation distance between two images $I_1$ and $I_2$ can be defined as the total variation norm of the difference $I_1 - I_2$, i.e., $TVD(I_1, I_2) = TV(I_1 - I_2)$. The above distance defines a pseudometric over the space of images as it satisfies the symmetry property and the triangle inequality, but may violate the identity of indiscernibles as an image obtained by adding the same value to all the pixel intensities has a distance of zero from the original image. However, as noted in section 4, our certificates hold even for this setting.

We certify total variation distance for the problems of dimensionality reduction and image reconstruction on MNIST [16] and CIFAR-10 [32]. The base-model for dimensionality reduction is an autoencoder that uses convolutional layers in its encoder module to map an image down to a small number of latent variables. The decoder applies a set of de-convolutional operations to reconstruct the same image. We insert batch-norm layers in between these operations to improve performance. For image reconstruction, the goal is to recover an image from small number of measurements of the original image. We apply a transformation defined by Gaussian matrix $A$ on each image to obtain the measurements. The base model tries to reconstruct the original image from the measurements. The attacker, in this case, is assumed to add a perturbation in the measurement space instead of the image space (as in dimensionality reduction). The model first reverts the measurement vector to a vector in the image space by simply applying the pseudo-inverse of $A$ and then passes it through a similar autoencoder model as for dimensionality reduction. We present results for $\epsilon_1 = 0.2, 0.4, \ldots, 1.0$ and $h = 2, 1.5$ and use 256 latent dimensions and measurements for these experiments in figure 5. To put these plots in perspective, the maximum TVD between two CIFAR-10 images could be $6 \times 31 \times 31 = 5766$ and between MNIST images could be $2 \times 27 \times 27 = 1458$ (pixel values between 0 and 1).

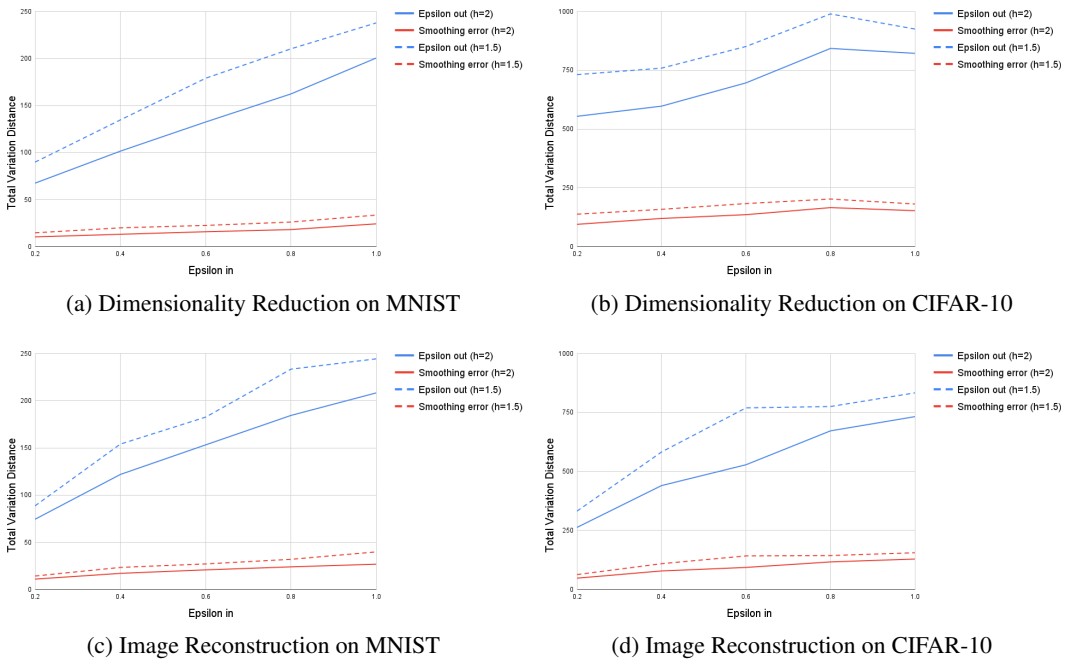

(a) Dimensionality Reduction on MNIST

(b) Dimensionality Reduction on CIFAR-10

(c) Image Reconstruction on MNIST

(d) Image Reconstruction on CIFAR-10

Figure 5: Certifying Total Variation Distance

# 6   Conclusion

Provable adversarial robustness can be extended beyond classification tasks to problems with structured outputs. We design a smoothing-based procedure that can make a model of this kind provably robust against norm bounded adversarial perturbations of the input. In our experiments, we demonstrate that this method can generate meaningful certificates under a wide variety of distance metrics in the output space without significantly compromising the quality of the base model. We also note that the metric requirements on the distance measure can be partially relaxed in exchange for weaker certificates.

We focus on $\ell_2$-norm bounded adversaries and the Gaussian smoothing distribution. An important direction for future investigation could be whether this method can be generalised beyond $\ell_p$-adversaries to more natural threat models, e.g., adversaries bounded by total variation distance, perceptual distance, cosine distance, etc. Center smoothing does not critically rely on the shape of the smoothing distribution or the threat model. Thus, improvements in these directions could potentially be coupled with our method to further broaden the scope of provable robustness in machine learning.

# 7   Acknowledgements

This work was supported by the AFOSR MURI program, DARPA GARD, the Office of Naval Research, and the National Science Foundation Division of Mathematical Sciences.

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
