# Center Smoothing: Certified Robustness for Networks with Structured Outputs

## Appendix

**Aounon Kumar**
University of Maryland
aounon@umd.edu

**Tom Goldstein**
University of Maryland
tomg@cs.umd.edu

## A    Proof of Theorem 2

Let $z' = \hat{f}(x')$. Then, by definition of $\hat{f}$,

$$\mathbb{P}\left[f(X') \in \mathcal{B}(z', \hat{r}(x', \Delta))\right] \geq \frac{1}{2} - \Delta, \tag{1}$$

where $X' \sim x' + \mathcal{P}$ and

$$\hat{r}(x', \Delta) = \min_{z''} r \text{ s.t. } \mathbb{P}[f(X') \in \mathcal{B}(z'', r)] \geq \frac{1}{2} + \Delta.$$

And, by definition of $\hat{R}$,

$$\mathbb{P}[f(X') \in \mathcal{B}(\hat{f}(x), \hat{R})] > \frac{1}{2} + \Delta. \tag{2}$$

Therefore, from (1) and (2), $\mathcal{B}(z', \hat{r}(x', \Delta))$ and $\mathcal{B}(\hat{f}(x), \hat{R})$ must have a non-empty intersection. Let, $y$ be a point in that intersection. Then,

$$d(\hat{f}(x), \hat{f}(x')) \leq d(\hat{f}(x), y) + d(y, z')$$
$$\leq \hat{r}(x', \Delta) + \hat{R}.$$

Since, by definition, $\hat{r}(x', \Delta)$ is the radius of the smallest ball with $1/2 + \Delta$ probability mass of $f(x' + \mathcal{P})$ over all possible centers in $\mathbb{R}^k$ and $\hat{R}$ is the radius of the smallest such ball centered at $\hat{f}(x)$, we must have $\hat{r}(x', \Delta) \leq \hat{R}$. Therefore,

$$d(\hat{f}(x), \hat{f}(x')) \leq 2\hat{R}.$$

## B    Proof of Lemma 1

Consider the smallest ball $\mathcal{B}(z', \hat{r}(x, \Delta_1))$ that encloses at least $1/2 + \Delta_1$ probability mass of $f(x + \mathcal{P})$. By Hoeffding's inequality, with at least $1 - e^{-2n\Delta_1^2}$ probability, at least half the points in $Z$ must be in this ball. Since, $r$ is the radius of the minimum enclosing ball that contains at least half of the points in $Z$, we have $r \leq \hat{r}(x, \Delta_1)$.

## C    Proof of Theorem 3

$\beta$-MEB$(Z, 1/2)$ computes a $\beta$-approximation of the minimum enclosing ball that contains at least half of the points of $Z$. Therefore, by lemma 1, with probability at least $1 - e^{-2n\Delta_1^2}$,

$$\beta\text{-MEB}(Z, 1/2) \leq \beta\hat{r}(x, \Delta_1) \leq \beta\hat{r}(x, \Delta),$$

35th Conference on Neural Information Processing Systems (NeurIPS 2021).

since $\Delta \geq \Delta_1$. Thus, the procedure to compute $\hat{f}$, if succeeds, will output a point $z \in \mathbb{R}^k$ which, with probability at least $1 - 2e^{-2n\Delta_1^2}$, will satisfy,

$$\mathbb{P}\left[f(X) \in \mathcal{B}(z, \beta\hat{r}(x, \Delta))\right] \geq \frac{1}{2} - \Delta.$$

Now, using the definition of $\hat{R}$ and following the same reasoning as theorem 2, we can say that,

$$d(\hat{f}(x), \hat{f}(x')) \leq \beta\hat{r}(x', \Delta) + \hat{R}$$
$$\leq (1 + \beta)\hat{R}.$$

## D  Proof of Lemma 2

Given $z = \hat{f}(x)$, define a random variable $Q = d(z, f(X))$, where is $X \sim x + \mathcal{P}$. For $m$ i.i.d. samples of $X$, the values of $Q$ are independently and identically distributed. Let $F(r)$ denote the true cumulative distribution function of $Q$ and define the empirical cdf $F_m(r)$ to be the fraction of the $m$ samples of $Q$ that are less than or equal to $r$, i.e.,

$$F_m(r) = \frac{1}{m} \sum_{i=1}^{m} \mathbf{1}_{\{Q_i \leq r\}}$$

Using the Dvoretzky–Kiefer–Wolfowitz inequality, we have,

$$\mathbb{P}\left[\sup_{r \in \mathbb{R}} (F_m(r) - F(r)) > \epsilon\right] \leq e^{-2m\epsilon^2}$$

for $\epsilon \geq \sqrt{\frac{1}{2m} \ln 2}$. Setting, $e^{-2m\epsilon^2} = \alpha_2$ for some $\alpha_2 \leq 1/2$, we have,

$$\sup_{r \in \mathbb{R}} (F_m(r) - F(r)) < \sqrt{\frac{\ln(1/\alpha_2)}{2m}}$$

with probability at least $1 - \alpha_2$. Set $r = \tilde{R}_q$, the $q$th quantile of of the $m$ samples. Then,

$$F(\tilde{R}_q) > F_m(\tilde{R}_q) - \sqrt{\frac{\ln(1/\alpha_2)}{2m}}$$

$$\text{or,} \quad \mathbb{P}\left[Q \leq \tilde{R}_q\right] > q - \sqrt{\frac{\ln(1/\alpha_2)}{2m}} = p.$$

With probability $1 - \alpha_2$,

$$\mathbb{P}\left[f(X) \in \mathcal{B}(\hat{f}(x), \tilde{R}_q)\right] > p.$$

## E  High-dimensional Outputs

For functions with high-dimensional outputs, like high-resolution images, it might be difficult to compute the minimum enclosing ball (MEB) for a large number of points. The smoothing procedure needs us to store all the $n \sim 10^3 - 10^4$ sampled points until the MEB computation is complete, requiring $O(nk')$ space, where $k'$ is the dimensionality of the output space. It does not allow us to sample the $n$ points in batches as is possible for the certification step. Also, computing the MEB by considering the pair-wise distances between all the sampled points is time-consuming and requires $O(n^2)$ pair-wise distance computations. To bring down the space and time requirements, we design another version (Smooth-HD, algorithm 1) of the smoothing procedure where we compute the MEB by first sampling a small number $n_0 \sim 30$ of candidate centers and then returning one of these candidate centers that has the smallest median distance to a separate sample of $n \ (\gg n_0)$ points. We sample the $n$ points in batches and compute the distance $d(c_i, z_j)$ for each pair of candidate center $c_i$ and point $z_j$ in a batch. The rest of the procedure remains the same as algorithm 1. It only requires us to store batch-size number of output points and the $n_0$ candidate centers at any given time, significantly reducing the space complexity. Also, this procedure only requires $O(n_0 n)$ pair-wise distance computations. The key idea here is that, with very high probability ($> 1 - 10^{-9}$), at least

**Algorithm 1** Smooth-HD

---

**Input:** $x \in \mathbb{R}^k, \sigma, \Delta, \alpha_1$.
**Output:** $z \in M$.
Set $C = \{c_i\}_{i=1}^{n_0}$ s.t. $c_i \sim f(x + \mathcal{N}(0, \sigma^2 I))$.
Set $\Delta_1 = \sqrt{\ln(2/\alpha_1)/2n}$.
Sample $Z = \{z_j\}_{j=1}^{n}$ s.t. $z_j \sim f(x + \mathcal{N}(0, \sigma^2 I))$ in batches.
For each batch, compute pair-wise distances $d(c_i, z_j)$ for $c_i \in C$ and $z_j$ in the batch.
Compute the center $c \in C$ with the minimum median distance to the points in $Z$.
Re-sample $Z$ in batches.
Compute $p_{\Delta_1}$.
Set $\Delta_2 = 1/2 - p_{\Delta_1}$.
If $\Delta < \max(\Delta_1, \Delta_2)$, discard $c$ and abstain.

---

one of the $n_0$ candidate centers will lie in the smallest ball that encloses at least $1/2 + \Delta_1$ probability mass of $f(x + \mathcal{P})$. Also, with high probability, at least half of the $n$ samples will lie in this ball too. Thus, the median distance of this candidate center to the $n$ samples is at most $2\gamma\hat{r}(x, \Delta_1)$, after accounting for the factor of $\gamma$ in the relaxed version of the triangle inequality as discussed in section 4. Ignoring the probability that none of the $n_0$ points lie inside the ball, we can derive the following version of theorem 3:

**Theorem 1.** *With probability at least $1 - \alpha_1$,*

$$\forall x' \text{ s.t. } \|x - x'\|_2 \leq \epsilon_1, \ d(\hat{f}(x), \hat{f}(x')) \leq \gamma(1 + 2\gamma)\hat{R}$$

*where $\alpha_1 = 2e^{-2n\Delta_1^2}$.*

## F   Baseline for $\ell_2$-Metric

In this section, we compare the certificates from center smoothing against a bound derived in [2] for functions like $f$ smoothed by taking the expectation of $f$ under a Gaussian noise. This bound only applies when the output metric is $\ell_2$. For a vector-valued function $f$, the change in the function defined as $\mathbb{E}_\delta[f(x + \delta)]$ where $\delta \sim \mathcal{N}(0, \sigma^2 I)$, under an $\ell_2$-perturbation of the input of size $\epsilon_1$, can be bounded by $(\max\|f\|_2 + \min\|f\|_2)\text{erf}(\epsilon_1/2\sqrt{2}\sigma)$. We apply our center smoothing procedure on the autoencoder and image reconstruction models used in section 6.3 with $\ell_2$ as the output metric and compare its certificates to the above bound. Since the minimum $\ell_2$-norm of the output of these models can be zero and we keep $h = \epsilon_1/\sigma = 2$ for these experiments, the change in the output of $\mathbb{E}_\delta[f(x+\delta)]$ can be bounded by $\max\|f\|_2\text{erf}(1/\sqrt{2}) \leq 0.68\sqrt{d}$, where $d$ is the number of dimensions of the output space. For $28 \times 28$ gray-scale MNIST images and $32 \times 32$ RGB CIFAR-10 images, the corresponding bounds are 19.04 and 37.69 respectively. Figure 1 shows that the certificates obtained for center smoothing remain below the baseline for all the values of $\epsilon_1$ used. Thus, by observing the neighborhood of an input point, center smoothing can yield better certificates for individual points in the input space than the baseline bound which is a global guarantee.

## G   Angular Distance

A common measure for similarity of two vectors $A$ and $B$ is the cosine similarity between them, defined as below:

$$\cos(A, B) = \frac{A \cdot B}{\|A\|_2 \|B\|_2} = \frac{\sum_i A_i B_i}{\sqrt{\sum_j A_j^2}\sqrt{\sum_k B_k^2}}.$$

In order to convert it into a distance, we can compute the angle between the two vectors by taking the cosine inverse of the above similarity measure, which is known as angular distance:

$$AD(A, B) = \cos^{-1}(\cos(A, B))/\pi.$$

Angular distance always remains between 0 and 1, and similar to the total variation distance, angular distance also defines a pseudometric on the output space. We repeat the same experiments with the

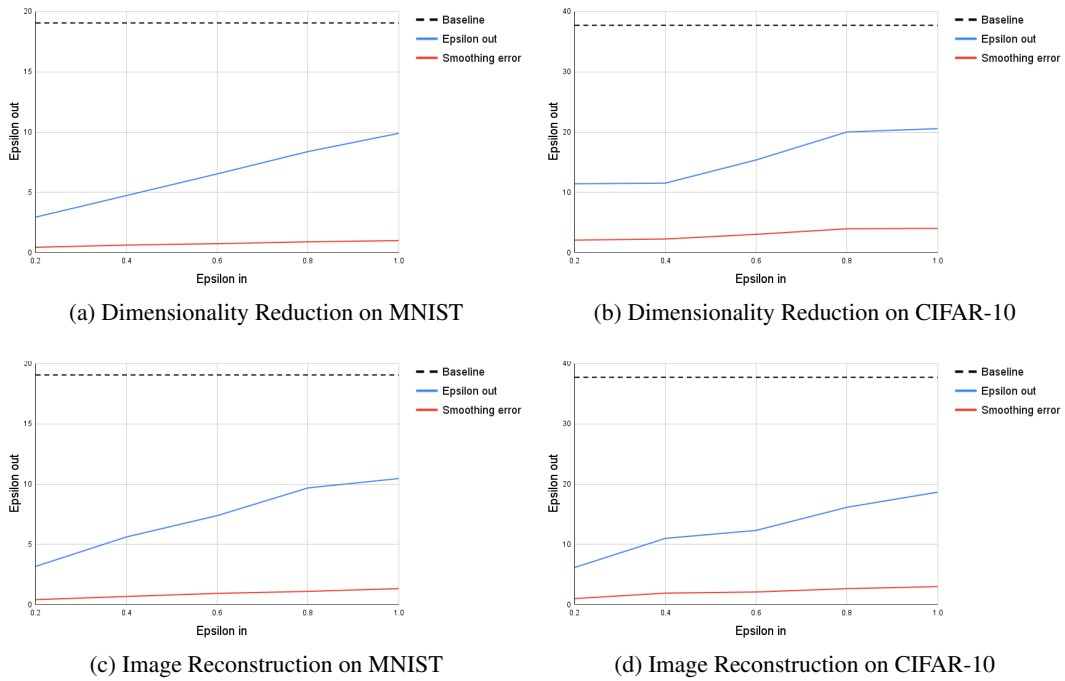

(a) Dimensionality Reduction on MNIST      (b) Dimensionality Reduction on CIFAR-10

(c) Image Reconstruction on MNIST      (d) Image Reconstruction on CIFAR-10

Figure 1: Comparison with baseline ($h = 2$).

same models and hyper-parameter settings as for total variation distance (figure 2). The results are similar in trend in all the experiments conducted, showing that center smoothing can be reliably applied to a vast range of output metrics to obtain similar robustness guarantees.

## H   Effect of Training with Noise

A common practice in the randomized smoothing literature is to train the base model with noise added to the training examples [1]. This helps the model to learn to ignore the smoothing noise and leads to better robustness certificates for classification tasks. For the total variation certificates in section 6.3, we train the autoencoders and the reconstruction models using a Gaussian noise with the same variance as the one used for prediction and certification. In this section, we perform an ablation experiment to study the effect of the training noise in the certified output radius of the base model (figure 3). We observe that both the smoothing error and the certified output radius deteriorate in the absence of training noise. However, models trained without noise also produce non-trivial certificates. This shows that both center smoothing and training with noise contribute towards the robustness and performance of the smoothed models.

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

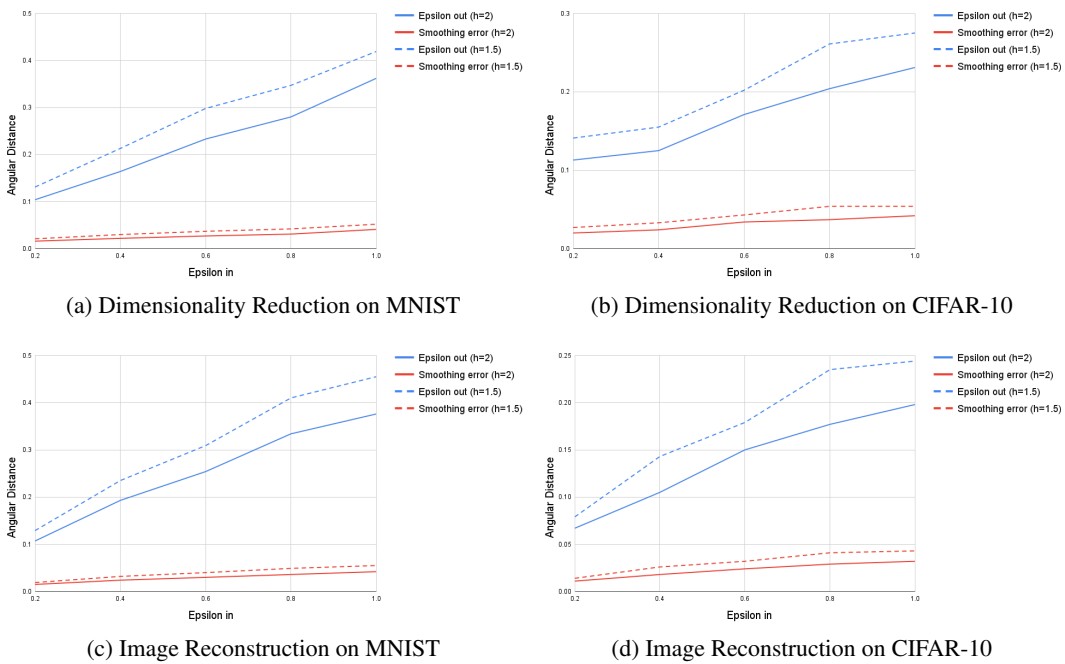

(a) Dimensionality Reduction on MNIST

(b) Dimensionality Reduction on CIFAR-10

(c) Image Reconstruction on MNIST

(d) Image Reconstruction on CIFAR-10

Figure 2: Certifying Angular Distance

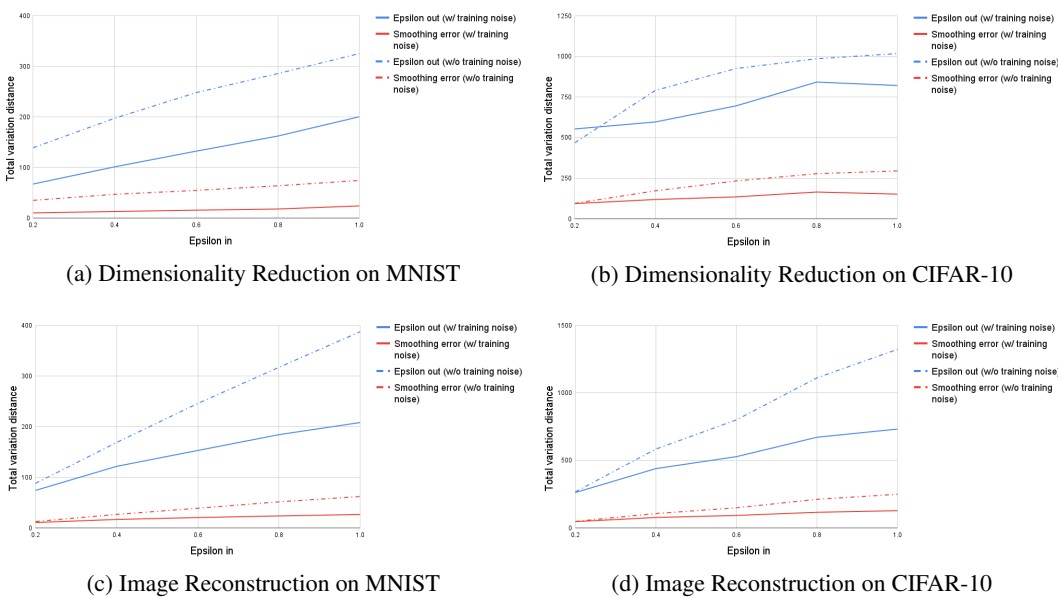

(a) Dimensionality Reduction on MNIST

(b) Dimensionality Reduction on CIFAR-10

(c) Image Reconstruction on MNIST

(d) Image Reconstruction on CIFAR-10

Figure 3: Impact of training noise on the performance of the robust model and its certificates.