# OpenReview forum: "Center Smoothing: Certified Robustness for Networks with Structured Outputs"
_NeurIPS.cc/2021/Conference — NeurIPS 2021 Poster_

### Official Review · Reviewer_aHRy · 2021-07-09

**Rating:** 5
**Confidence:** 4

**Summary:**

The paper proposes an extension to randomized smoothing to certify robustness of functions over more general metric spaces by finding the center of the smallest ball which contains at least half of the output of a set of points. They approximate this center by finding the point with the smallest median distance to the other points.

**Limitations And Societal Impact:**

Sure

**Main Review:**

I think this paper feels a bit like a solution in search of a problem.

I've seen a lot of work take randomized smoothing and do some small change to apply it to a slightly different setting, without presenting much work that seems truly novel. These sorts of works are worth sharing, but I don't personally feel they're worth publishing individually at a top conference. I don't believe that smoothing over general metric spaces is useless, and of course I know that the authors had to do some work to bridge the gap from A to B---hence I feel the contribution does have merit. But ultimately this paper presents a very minor development necessary to apply randomized smoothing to [slightly new setting X that no one has yet staked a claim on].

The presentation is decent. The experiments are decent. The math is trivial, but fine. This is the only meaningful summary I feel I can give on this work: it is technically correct, but not insightful and the math is trivial. So I guess the question is, what is the expectation for an accepted publication? This is a subjective question. I think it should be more than just a stamp of approval that the math is technically correct, so while I don't think this work is bad, I would not recommend publication.

Additional notes:

* Line 39 is not correct, randomized smoothing was not first studied by Cohen et al. The idea of injecting noise a la differential privacy is due to [1], and the formulation in its current form was first suggested in [2].

* Are the authors aware that smoothing a bounded function with Gaussian noise already naturally induces certified robustness in the form of Lipschitzness? This is known as the Weierstrass transform, see Appendix A of [3]. I'm not 100% sure but I think this could serve as a (much faster, simple) baseline to compare when evaluating bounded functions such as IOU?

[1] Certified Robustness to Adversarial Examples with Differential Privacy. Lecuyer et al. 2018

[2] Certified Adversarial Robustness with Additive Noise. Li et al. 2019 (this paper was originally called "Second-Order Adversarial Attack and Certifiable Robustness" and was first posted to arXiv in 2018).

[3] Provably Robust Deep Learning via Adversarially Trained Smoothed Classifiers. Salman et al. 2019


---------------------
**Update post rebuttal**

The authors and I had some fruitful discussion. However, my feeling that the novelty and significance of the submission are below the bar for publication remains. I think this is an ok idea, but the work itself just isn't a paper's worth of material. I encourage the authors to further explore this idea; fleshed out with additional experiments, more finetuned theoretical analyses, or other improvements I expect this could be a reasonable publication.

**Time Spent Reviewing:**

2.5

---

> ### Author Response · Authors · 2021-08-10
> **Novelty, Significance and Other Concerns**
>
> Below we address the concerns raised in the main review:
>
> 1. **Novelty and significance:** We differ from the reviewer's view that the setting considered in our work is only slightly different from what has been previously studied.
>     Most provable robustness methods developed so far focus on certifying the output of a classifier or a one-dimensional real-valued output.
>     However, these types of problems are only a minority in real-world applications of deep neural networks (DNNs).
>     For example, DNNs are used to detect objects/faces where the output is a set, perform segmentation of an image where the output is a labelling of each pixel, summarize a scene/picture where the output is natural language, produce an image from latent representation (in GANs), control robots where the output is the angle, position, velocity, etc. of different parts, drive a vehicle where the output is direction, acceleration, velocity, etc., and produce medical diagnoses from scans, each of which is difficult to capture as a classification task or one that outputs a one-dimensional real-valued quantity.
>     The outputs are often abstract objects such as sets, language, distributions, or a long sequence of real numbers as in images.
>     In these settings, the individual components of the output (like pixels in an image) are less important than the overall meaning/semantics of the entire output with all its components taken together.
>     Adversarial attacks on these models have also been extensively studied, e.g., references [1, 5, 6, 48, 45, 13, 40, 8] and [52] in our paper, to list only a few of them.
>     For attacks where, say, an adversary tries to make an object disappear from a segmentation output or tries to insert malicious artefacts in the reconstruction of an MRI scan to influence a medical diagnosis, it is necessary to first formulate what the goal of adversarial robustness should be before we can design procedures to defend against them.
>
>     In our work, we first address this question: what is the right notion for a certificate in these settings?
>     Intuitively, it makes sense for any robustness procedure to preserve the output of the base model, e.g., the class outputted by a classifier.
>     But, in the aforementioned scenarios, this question becomes difficult to answer.
>     For instance, do we want the value of every single pixel in an image or segmentation output to remain the same? This might be too strong a requirement to have and may not always lead to good and meaningful certificates.
>     Instead, we can allow the pixels to change as long as the overall output does not differ significantly from the original (no attack) output in its meaning. This difference in meaning is measured by the distance function $d$ which could be perceptual distance, TVD, etc., for image outputs, IoU for sets, the fraction of disagreeing pixels for segmentation outputs, etc.
>
>     We then show a randomized-smoothing based procedure to achieve the type of certificates described above with minimal assumptions on the distance function $d$. (The only crucially requirement is for $d$ to satisfy the triangle inequality, that too only approximately!)
>     Some recent works have attempted to generate certificates in these settings by considering some statistical information about the individual components of the output. However, they are specific to the distance measures considered and are difficult to extend to other distance functions, e.g., it is hard to see why something like the mean/median of the individual pixels be robust under a perceptual distance function like LPIPS (a metric considered in one of our experiments included in the appendix).
>     Our method has the distance function baked into the smoothed model and is oblivious to the inner workings of the distance function.
>     Center smoothing significantly extends the scope of provable adversarial robustness by generalising randomized smoothing to a vast class of problems beyond classification tasks (classification can also be cast into our framework using a distance function which outputs 0 or 1 based on whether the output classes are the same or not respectively).
>
> 2. **Mathematical complexity and insight:** Our work presents a simple and easy way to provably defend against a much wider range of adversarial attacks than what existing methods can achieve.
>     We present a short and easy-to-follow proof of why the theoretical smoothed function $\bar{f}$ (center of the high density ball in the output space) is certifiably robust in the metric setting.
>     The main objective of including this proof was to give the reader the intuition behind the working principle of center smoothing.
>     A key insight here is that randomized smoothing allows us to achieve robustness in a metric-agnostic fashion.
>     Although the theorem is simple and its proof easy to understand, it is not so straightforward to come up with its right formulation.
>     For instance, why should one consider returning the center of the smallest ball that encloses half the probability mass?
>     It is inspired from the 1-center with outliers problem studied extensively (along with its many variants and generalizations) in theoretical computer science.
>
>
>     Moreover, the theoretical smooth function $\bar{f}$ is not easy to compute. In fact, solving the 1-center problem is NP-hard.
>     Plus, the output distribution is unknown and we may only draw i.i.d samples from it.
>     To get around these hurdles, we defined multiple approximations to $\bar{f}$ and showed that all of them were provably robust which finally yields the desired robustness certificate.
>     In addition, we modify the smoothing algorithm (alg. Smooth-HD) in the appendix to handle functions with very high-dimensional outputs where the space required to store the output samples needed to compute the center is too high.
>     We show that it is sufficient to sample a small set ($\sim 30$) of potential centers and return the one with the smallest median distance to a set of separately sampled points to achieve the same robustness guarantees as before.
>     This lets us process the sampled output points in batches, which significantly reduces the storage requirements of the smoothing step.
>     Considering all the steps involved in the process for achieving the final robustness goal, we do not believe it can be easily dismissed as being trivial.
>
> 3. **References:** Indeed, Lecuyer et al. and Li et al.'s works precede Cohen et al.'s on arXiv. The conference versions we cited are all from 2019. Thank you for pointing this out. We will correct it in the updated version.
>
> 4. **Bounded output:** We are aware of the result that shows Lipschitzness of a bounded function under Gaussian smoothing noise. However, this result does *not* apply to our setting because even though IoU is bounded between 0 and 1, it is *not* the output of the function being smoothed.
>     The output is a set. The distance function $d$ (e.g., IoU) measures how much the output changes under an adversarial perturbation of the input.
>     The output of the base function in our case could be an abstract entity such as image, probability distribution, graph node, set, etc., which cannot be treated as a bounded real value in a meaningful way.

---

> > ### Comment · Reviewer_aHRy · 2021-08-12
> > **Still not sure**
> >
> > Thanks for your thorough response. I did not mean to imply in my review that adversarial examples are not worth studying, nor that robustness of general metric-output functions is not important. Rather, adversarial examples are somewhat of an "academic" field of study---they are very meaningful as a formally defined problem to work to solve. On the other hand, they're not really a "real world" problem. That's not to say they don't apply in the real world, but it's not something that practitioners seriously give thought to, as far as I know. If they do, I imagine they just perform adversarial training or use a pretrained robust model. Certified robustness of DNNs, while critical as a direction of study, isn't really a real-world issue at the moment.
> >
> > Why is this distinction important? Here's a question to think about (though no need to actually answer): when beginning this research direction, what was the primary motivating factor? From what I can tell, there are two possibilities:
> >
> > 1. "General metric-output DNNs are perilously non-robust, and this is a real problem! What solutions could we consider to overcome this?"
> >
> > 2. "Randomized smoothing is a cool algorithm, with lots of applications. What hasn't it been used for already, such that we can modify it and apply it to that problem?"
> >
> > This paper feels like an instance of number 2. Obviously, the original motivation for a project should not be a factor when judging its merit. The point I'm trying to convey is that, to me, the "novelty" of this work appears to be simply the *decision* to try to adapt randomized smoothing to this slightly different setting. The algorithm, while not trivial, is essentially exactly what one would expect (I'm being very careful here---it is often the case that something which appears obvious "in hindsight" may have been quite complex to derive. I feel quite strongly that this is *not* the case in this instance. The algorithm is exactly the "Neyman-Pearson lemma" idea behind randomized smoothing, changed for metric outputs). The math is totally trivial. And the experiments are pretty much the minimum acceptable level of thoroughness. I could perhaps imagine a paper that solves *multiple* of these "adaptations" being a full-fledged publication, but this one idea alone does not, in my opinion, meet the bar for novelty or significance.
> >
> > I'm aware that this response comes off as a bit dismissive. I want to emphasize that I don't think this is a bad contribution---I just don't see it as being at the level of significance or novelty I expect in order to recommend for publication.
> >
> > * **Re: bounded output.** I realize that the function being smoothed outputs a set, and therefore this would not directly apply. But couldn't you just as easily define the *composition* of the function and the IOU evaluation as the base function that is being smoothed? When training, you only train the function which outputs a set, but at evaluation time, I believe this is a perfectly legitimate way to certify the base function. Because ultimately, the *reason* you're certifying the output of the function is to show that the IOU will not decrease by too much, isn't that correct? Am I still missing something here?

---

> > > ### Author Response · Authors · 2021-08-12
> > > **Re: Still not sure**
> > >
> > > **Bounded Output:** It is not possible to compose the distance function $d$ (1-IoU) with the base function $f$ as the base function takes an input $x$ and outputs just one element in the output space, say, $y = f(x)$, but the distance function $d$ requires two elements $y_1$ adn $y_2$ in the output space as inputs to compute the distance $d(y_1, y_2)$ between these elements.
> > > The guarantee of the certificate is that for a given input $x$, the distance $d(\bar{f}(x), \bar{f}(x'))$ is bounded by $\epsilon_2$ for any perturbed version $x'$ of $x$ as long as $||x - x'||_2 \leq \epsilon_1$ (Please refer to line 50 of the paper).
> > > When we are certifying an input $x$, we do not have $x'$ available to us and so, we cannot compute $d(\bar{f}(x), \bar{f}(x'))$ directly.
> > >
> > > For the case of IoU, the certificate would say: for a given input $x$, $1 - IoU(\bar{f}(x), \bar{f}(x')) \leq \epsilon_2$ for any $x'$ such that $||x - x'||_2 \leq \epsilon_1.$
> > > The certificate does not say "IOU will not decrease by too much."
> > > In fact, there is no IoU to begin with!
> > > The certificate says the IoU (Intersection over Union, which is a measure of the overlap of two sets) between the output $\bar{f}(x)$ for the original input $x$ and the output $\bar{f}(x')$ for the perturbed input $x'$ is guaranteed to be above a certain threshold (say 80\%) for any $x'$ that satisfies $||x - x'|| \leq \epsilon_1$.
> > > The IoU of just the output $\bar{f}(x)$ is undefined as it requires two inputs to compute the distance between them.
> > >
> > > We are not trying to certify a bounded real-valued function of the output of the base function.
> > > That would be trivial! It is not possible to cast the problem of certifying metric-space outputs as certifying a composition of the base function and the distance function where you could directly apply the Weierstrass transform.
> > > Your concerns regarding the novelty, significance and motivation of our work appear to have stemmed from a lack of understanding of the problem we are solving and the procedure we have developed.
> > > Perhaps spending a little more time on the paper and going a bit more into the technical details will help you understand our work better and see its true impact.

---

> > > > ### Comment · Reviewer_aHRy · 2021-08-12
> > > > **I think you might still misunderstand me here**
> > > >
> > > > "Your concerns regarding the novelty, significance and motivation of our work appear to have stemmed from a lack of understanding of the problem we are solving and the procedure we have developed."
> > > >
> > > > Obviously no one can say this with 100% certainty, but I'm pretty sure this is not the case. I am intimately familiar with randomized smoothing, and this discussion is on a side-remark I made without thinking too deeply about it. My understanding of the paper is not related to the correctness of the tentative suggestion I made (which I acknowledged from the beginning could be wrong).
> > > >
> > > > However, I think you're still not understanding what I'm suggesting here. Here's what you just wrote:
> > > > "The certificate says the IoU (Intersection over Union, which is a measure of the overlap of two sets) between the output $\bar f(x)$
> > > >  for the original input $x$ and the output $\bar f(x')$ for the perturbed input $x'$ is guaranteed to be above a certain threshold (say 80%) for any $x'$ that satisfies $||x-x'|| \leq\epsilon_1$."
> > > >
> > > > Let's unpack this. The formal statement is:
> > > > $$\forall x'\in\mathcal{B}_{\epsilon_1}(x).\ 1-IOU(\bar f(x), \bar f(x')) \leq \epsilon_2.$$
> > > >
> > > > Trivially, this is **equivalent** to:
> > > > $$\forall x'\in\mathcal{B}_{\epsilon_1}(x).\ IOU(\bar f(x), \bar f(x')) \geq 1-\epsilon_2.$$
> > > >
> > > > What is this second statement saying? I contend it's saying *exactly* what I claimed: for all $x'$ in a certain radius of the original input $x$, the IOU of the perturbed output (with the "reference output" $\bar f(x)$) will not decrease by too much (i.e., more than $\epsilon_2$). This is because the IOU of a set with itself is 1. In other words, at the original input $x$, we have that the IOU is 1. Now, for any given perturbation $x' := x + \delta$, how much can this change the IOU? If the *composition* of $f$ and the IOU evaluation is smoothed, then it is $L$-Lipschitz for some $L$. This means you can give the same guarantee for $\epsilon_2 = \epsilon_1 / L$. In other words,
> > > >
> > > > $$1 - IOU(\bar f(x), \bar f(x')) = |IOU(\bar f(x), \bar f(x)) - IOU(\bar f(x), \bar f(x'))| \leq L ||x - x'||$$
> > > >
> > > > Thus you have a simple baseline which is trivial to compute.
> > > >
> > > > Does this make sense?

---

> > > > > ### Author Response · Authors · 2021-08-12
> > > > > **Composition of base function and distance function**
> > > > >
> > > > > "If the composition of $f$ and the IOU evaluation is smoothed" -- Could you please clarify what the composed function is?
> > > > > In order to apply the Weierstrass transform, we first need a function of the form $g: \mathbb{R}^n \rightarrow [0, 1]$ which maps the input space to a bounded interval.
> > > > > We would then apply the transform to obtain the smoothed version $\bar{g}$ which is the expectation under the Gaussian smoothing distribution,
> > > > > which would be $L$-Lipschitz for some $L$.
> > > > > How are you defining $g$ (the composition) in your suggestion?
> > > > > Also, note that smoothing the composition, if it can be defined, this way does not yield a smoothed version of $f$.
> > > > > The output of the smoothed function $\bar{f}$ has to be of the same type as $f$ (a set, in this case).
> > > > > But the output of the smoothed version of the composition would be a real value between 0 and 1.
> > > > > How do we get a set from this value?

---

> > > > > > ### Comment · Reviewer_aHRy · 2021-08-12
> > > > > > **I think you're right**
> > > > > >
> > > > > > Alright, I've given this some more thought and I think you're right that this would not work. Here's vaguely what I had in mind:
> > > > > >
> > > > > > Given a trained function $f$, at test time we predict $\bar f(x) =: \hat y = E_{p(\epsilon)}[f(x+\epsilon)]$.
> > > > > >
> > > > > > Next, define $g(z) = IOU(\hat y, \bar f(z)) = IOU(\hat y, E_{p(\epsilon)}[f(z+\epsilon)])$.
> > > > > >
> > > > > > Before writing this down formally, my thought was to essentially pull the expectation outside the IOU, such that $g$ would be Lipschitz and then one could certify that $IOU(\hat y, \bar f(z))$ is small whenever $||x-z||$ is small. But after writing it out it's clear that by pulling out the expectation, we are no longer evaluating IOU on the smoothed function, so this doesn't make sense. Your note that "smoothing the composition, if it can be defined, this way does not yield a smoothed version of $f$" is essentially that same point.
> > > > > >
> > > > > > Sorry for the confusion!

---

> > > > > > > ### Author Response · Authors · 2021-08-12
> > > > > > > **A Baseline**
> > > > > > >
> > > > > > > Thank you for the clarification! We really appreciate you taking the time to engage with us.
> > > > > > > We will include a brief summary of our conversation in the next version of our paper to motivate the necessity of our method.
> > > > > > >
> > > > > > > In a previous version of this paper, we did compare center smoothing to a baseline certificate which is somewhat similar to the one you had in mind but only holds when the output metric is the $\ell_2$-norm of the difference of the outputs, i.e., $||\bar{f}(x) - \bar{f}(x')||_2$ (please refer to notes on submission history if they are visible to you).
> > > > > > > We compared our certificates to the one in the article: Making medical image reconstruction adversarially robust by Adva Wolf.
> > > > > > > This paper derives a Lipschitz-bound for a smoothed function defined as the element-wise mean of the output vectors when the input is sampled from a Gaussian smoothing distribution.
> > > > > > > Our certificates always outperformed this baseline by a significant margin.
> > > > > > > We removed those results to include more interesting output distance metrics like TVD, Jaccard distance, perceptual distance (LPIPS), etc., but we can include those back in the next update of our paper.
> > > > > > > Below we present some of those results ($h = \frac{\epsilon_1}{\sigma} = 2$):
> > > > > > >
> > > > > > > **Dimensionality Reduction (CIFAR-10):**
> > > > > > >
> > > > > > > | Epsilon in                 |    0.2 |    0.4 |    0.6 |    0.8 |      1 |
> > > > > > > |----------------------------|-------:|-------:|-------:|-------:|-------:|
> > > > > > > | Lipschitz bound (baseline) | 37.839 | 37.839 | 37.839 | 37.839 | 37.839 |
> > > > > > > | Epsilon out                | 11.438 | 11.546 | 15.385 |  20.02 | 20.569 |
> > > > > > > | Smoothing error            |  2.103 |   2.29 |  3.062 |  3.982 |  4.043 |
> > > > > > >
> > > > > > > **Dimensionality Reduction (MNIST):**
> > > > > > >
> > > > > > > | Epsilon in                 |     0.2 |     0.4 |     0.6 |     0.8 |       1 |
> > > > > > > |----------------------------|--------:|--------:|--------:|--------:|--------:|
> > > > > > > | Lipschitz bound (baseline) | 19.1156 | 19.1156 | 19.1156 | 19.1156 | 19.1156 |
> > > > > > > | Epsilon out                |   2.958 |   4.743 |   6.537 |   8.384 |   9.896 |
> > > > > > > | Smoothing error            |   0.458 |   0.641 |   0.757 |   0.909 |   1.012 |
> > > > > > >
> > > > > > > **Image Reconstruction (CIFAR-10):**
> > > > > > >
> > > > > > > | Epsilon in                 |    0.2 |    0.4 |    0.6 |    0.8 |      1 |
> > > > > > > |----------------------------|-------:|-------:|-------:|-------:|-------:|
> > > > > > > | Lipschitz bound (baseline) | 37.839 | 37.839 | 37.839 | 37.839 | 37.839 |
> > > > > > > | Epsilon out                |  6.188 | 11.012 | 12.328 | 16.157 | 18.662 |
> > > > > > > | Smoothing error            |  1.037 |  1.938 |  2.128 |  2.678 |  3.015 |
> > > > > > >
> > > > > > > **Image Reconstruction (MNIST):**
> > > > > > >
> > > > > > > | Epsilon in                 |     0.2 |     0.4 |     0.6 |     0.8 |       1 |
> > > > > > > |----------------------------|--------:|--------:|--------:|--------:|--------:|
> > > > > > > | Lipschitz bound (baseline) | 19.1156 | 19.1156 | 19.1156 | 19.1156 | 19.1156 |
> > > > > > > | Epsilon out                |   3.182 |   5.622 |   7.387 |   9.682 |  10.459 |
> > > > > > > | Smoothing error            |   0.429 |   0.691 |   0.948 |   1.112 |   1.342 |
> > > > > > >
> > > > > > > Observations: Since $\frac{\epsilon_1}{\sigma}$ is kept constant, the Lipschitzness-based certificate (baseline) also remains constant (refer to theorem 1 of the above article).
> > > > > > > The above tables show that the certified output radius produced by center smoothing always remains below the baseline and thus is tighter (lower is better).

---

### Official Review · Reviewer_ujD6 · 2021-07-12

**Rating:** 6
**Confidence:** 3

**Summary:**

The authors extends randomized smoothing, originally developed for classifiers, to functions which map inputs to a general metric space. The extension seems intuitive but is technically challenging. The authors develop tractable, statistical methods both for finding the smooth prediction as well as computing the certificates. Various experiments are conducted to highlight its flexibility in different metrics and scenarios, such as Jaccard distance w.r.t. face detection, total variation distance w.r.t. dimension reduction, and angular distance for image reconstruction.

**Limitations And Societal Impact:**

There seems no immediate negative societal impacts. The authors could also discuss potential methods that would improve the tightness of their method.

**Main Review:**

1. Well-executed statistical methods for deriving and computing the certificates of robustness. It is not clear how tight the certificate is, though.

2. It is not clear to me how to choose $\Delta$. How exactly to select $\Delta_1$ appropriately so the process succeeds easily (as written in line 161)? Can the user optimize $\Delta$ to improve the certificate (given the same smoothed model)?

3. I think line 107 and the first part of Section 3.2 can be written in a much more clear way by writing the "classifier" down explicitly. For example, you can write it as something like $P(\mathbb{I}(f(x + \delta) \in B))$ and it will be clear that $\mathbb{I}(f(x + \delta) \in B)$ is a classifier so you can apply the result from (Cohen et al. 2019).

4. Theorem 1 is very clean, highlighting how the certificate is derived in a high-level, ideal scenario, but since it is very similar to Theorem 2 and Theorem 1 is not really used, I wonder whether it is possible to simplify this part.

5. The derivation of the certificate is not very clean, with some details missing / hiding in the Appendix. In the Appendix, since there is no page limitation, I suggest the authors to write every step in detail so as to make this paper self-contained. For example, first writing Hoeffding's inequality and DKW inequality down and then showing how the inequalities are applied can help the readers understand the proof.

6. The problem is not strongly motivated. It is not clear how severe current systems would fail and what the consequences would be in the context considered in this paper. For example, how non-robust can a model without smoothing behaves for dimension reduction? Even if it is indeed not very robust, what would be the real world consequence? Based on the writing, the authors only indicate the existence of adversarial examples and motivate robustness for face recognition, but it is not really clear how bad existing methods are.

In contrast, it is very clear how bad and imperceptible adversarial example can be in image classifiers, and the consequence is trivial since many mission-critical problem (e.g., face identification) are formulated as classification problems. The contribution of this paper is currently very limited to its technical discovery due to this reason.

7. In addition, there is no empirical comparison to existing models. Even though this work might be the first work for certified robustness w.r.t. metric space outputs, the authors can still compare it with vanilla models without smoothing, and show that the vanilla model would fail (i.e. finding an adversarial example) under the certified region given by the smoothed model. Clearly the authors can run simple gradient methods to find adversarial examples for total variation distance and angular distance.

8. Lots of space are wasted for large figures in experiment sections and simple concepts (e.g., how cosine is computed). I suggest the authors to shrink figure sizes and replace trivial concepts with technical details in the proofs to make the method more understandable.

9. Nevertheless, the paper still seems technically solid (if the proof are indeed correct), and the problem seems new and interesting (although the significance is questionable), I will recommend this paper for marginally above the acceptance threshold.

Minor:
In line 103 and 104, I think it should be $h(x + \delta)$ and $h(x' + \delta)$.

Disclaimer:
I did not read all the proofs in the Appendix.

**Time Spent Reviewing:**

6

---

> ### Author Response · Authors · 2021-08-10
> **Tightness of Certificate, Choosing Parameters and Problem Motivation**
>
> Below we address the concerns raised in the review:
>
> 1. "Well-executed statistical methods for deriving and computing the certificates of robustness. It is not clear how tight the certificate is, though." -- Our analysis is tight for a general distance function $d$.
>     The bound of $2R$ in theorem 1 will be tight when the two balls in fig. 2 are touching each other at just one point and the centers are both distance $R$ away from this point.
>     Similarly the 2-approximation of the minimum enclosing ball can be shown to be tight for a suitable distance metric, such as, graph distance.
>     The bounds can be improved, however, for specific distance functions, e.g., there are better approximations for the minimum enclosing ball for the $\ell_2$-distance metric as we discuss in lines 176 and 177 in page 5.
>     The tightness in the worst-case drop in the probability mass inside the ball $\mathcal{B}(\hat{f}(x), \hat{R})$ for a perturbation of size $\epsilon_1$ of the input follows from the tightness of the Gaussian smoothing certificate in eq. (1).
>     Thus, the $p$ calculate in eq. (4) is the minimum probability that the ball must enclose so that the probability mass inside it does not fall below $1/2 + \Delta$ in the worst-case for an $\epsilon_1$-size perturbation of the input $x$. We will include this discussion in the next update of the paper.
>
>
> 2. "It is not clear to me how to choose $\Delta$." -- The value of $\Delta_1$ depends on $\alpha_1$ (and $n$, see thm. 3) which is one of the components of $\alpha = \alpha_1 + \alpha_2$ the overall failure probability of the certificate.
>     $\Delta_2$ is computed as described in lines 156 - 159.
>     Finally, $\Delta$ is set to some value such that it satisfies $\max(\Delta_1, \Delta_2) \leq \Delta$ for most of the experiments.
>     This step is done through empirical observation of how often the smoothing step abstains.
>     We want to keep $\Delta$ as small as possible (as it gets better certificates) without violating $\max(\Delta_1, \Delta_2) \leq \Delta$ too often. In our experiments, we ensure that the procedure does not abstain for more than 10% of the examples.
>     To reduce $\Delta_1$ and $\Delta_2$ for the same failure probability $\alpha$, we will need to increase the number of samples $n$ required to compute the center.
>
>
> 3. **Problem motivation and real-world attacks:** Adversarial attacks in the setting we considered have been studied extensively in the literature. Almost any application of deep neural networks faces the threat of adversarial attacks.
>     We list some of these in lines 22 - 28 which include applications like reconstruction of MRI scans and generating super-resolution images along with facial recognition.
>     Generative models have also been shown to be vulnerable to adversarial attacks (Adversarial examples for generative models by Kos et al. 2017) and we have included additional experiments in the appendix where we certify the output of a pre-trained GAN that outputs high-dimensional ImageNet images.
>     Recently, we have also come across adversarial attacks on segmentation models and image-to-language models like image captioning, e.g., Adversarial Examples for Semantic Image Segmentation by Fischer et al. 2017, Adversarial Examples for Semantic Segmentation and Object Detection by Xie et al. 2017 and Attacking Visual Language Grounding with Adversarial Examples: A Case Study on Neural Image Captioning by Chen et al. 2018.
>     Segmentation models can be certified by center smoothing using a distance function that returns the fraction of pixels where two segmentation outputs differ. Similarly, interesting distance functions that capture the semantic meaning of natural language can be considered for certifying image-to-language models.
>
>    Real-world attacks do not directly attack the distance function in these settings. Instead, they seek to change the output in such a way that will influence the outcome of some task that depends on the output.
>     It requires domain knowledge to construct the right objective function to apply gradient methods in order to produce a successful attack.
>     For instance, an attacker might seek to introduce/erase artefacts (e.g., a tumor) from a reconstructed MRI scan to alter the medical diagnoses that rely on the reconstructed scan.
>     Or, an attacker could make a segmentation/detection output not recognize humans on the street, which would be a safety concern in a self-driving vehicle that uses the output of these models to make split-second decisions.
>     However, it is not immediately obvious how to characterize the change in the output in order to design robustness procedures for such problems.
>     For instance, seeking to certify the label of each and every pixel of a segmentation output might be too strong a requirement.
>     A recent work Scalable Certified Segmentation via Randomized Smoothing by Fischer et al. 2021 addresses this issue by leaving some pixels in the output unlabelled leading to segmentation outputs that have white spaces in them.
>     The smoothed model from center smoothing, in this case, will output a valid segmentation output with a certificate that will guarantee that the fraction of pixels that any adversary can change with an $\epsilon_1$-size perturbation of the input image will not exceed 15\% (say).
>
> 4. **Suggestions for improvements:** We thank the reviewer for the various suggestions on re-writing key sections of the paper to make it more understandable and self-contained.
>     We will take all of them under consideration in the next update of our paper.

---

> > ### Comment · Reviewer_ujD6 · 2021-09-10
> > **Feedback**
> >
> > I agree that some existing models do have terrible adversarial examples in some scenarios (e.g., segmentation) based on the cited papers. However, there are many problems in this paper relating to this.
> > 1) Re my point 6, it is not clear how bad the existing models are. The paper and the rebuttal only suggest that there are many paper studying this problem without indicating how bad the situation could be. For example, it could be that X% of the data has a "bad" adversarial example, but the readers will not have an idea about the value of X and the "badness".
> > 2) In addition, since the authors have cited many works that highlight the existence of adversarial examples in these metric spaces, why did the author not compare to any one of them, qualitatively or quantitatively? (Although probably not true, it could be the case that the architectures used by the authors are already robust without smoothing, right?) This echos my previous Main Review point 7, which is not addressed by the rebuttal.
> >
> > If the authors could highlight that a model without smoothing would fail dramatically with, e.g., real adversarial examples and their segmentations, while showing that adding the proposed extra layer of smoothing could improve the robustness dramatically with verifications, it will make this paper so much stronger. I still think this paper is only marginally above the acceptance threshold.

---

> > > ### Author Response · Authors · 2021-09-10
> > > **Re: Feedback**
> > >
> > > Thank you for your response! Below we address the concerns raised:
> > >
> > > 1. Attacks developed by previous works, for problems in our setting, work for almost all the points in the dataset and can make the model perform poorly on all of them. Audio Adversarial Examples: Targeted Attacks on Speech-to-Text by Carlini and Wagner is another example of adversarial attacks on a model that is not a classifier. As they show in their work, the attacks work with 100% success and can change the model’s output to a given output. Our method can produce robustness certificates under a desired distance metric for the output text, such as semantic distance, Jaccard distance, edit distances, etc. Recently, a perceptual hash function used by Apple in their CSAM detection system called NeuralHash was attacked using adversarial examples. It was shown that by adding an imperceptible perturbation to an image it is possible to drastically change its hash. Perceptual hash functions work by mapping similar looking images to vectors that are close to each other and perceptually different images to vectors that are far apart. In neural network based hashes, these vectors are generated using the high-level features produced by a trained network and are then converted to strings of integers using a technique called Hyperplane Locality Sensitive Hashing. Thus by changing the feature vectors one can alter the final hashes. The similarity between the feature vectors is measured using metrics such as cosine similarity, angular distance, etc. The attacks on NeuralHash sought to find an adversarial example whose feature vectors would have a high dissimilarity (say, by minimizing the cosine similarity) with the feature vector of the clean image.
> > >
> > > 2. We design a perceptual hash by taking the high-level features from a pre-trained ResNet18 model applied to 224 x 224 ImageNet images. We perform a PGD attack on this model and show that the output of the model can be very different for a clean image and an adversarial image in terms of cosine similarity. We applied center smoothing to certify these feature vectors under the cosine distance (1 - cosine similarity) metric. A cosine similarity of one means that the two vectors are completely aligned with each other and a similarity of negative one means that they are pointing in opposite directions. Note that the cosine distance does not satisfy the triangle inequality, but satisfies an approximate version of it with $\gamma = 2$. However, center smoothing can generate certificates even for such distance measures (see section 4 page 6). Below we report the median certified cosine similarity of the robust model, the median smoothing error (cosine distance between outputs of the base model and the robust model), the adversarial cosine similarity for the base model and the robust adversarial similarity (cosine similarity between clean and adversarial inputs) for the smoothed model computed for fifty randomly sampled images from the dataset for different input radii. We use $h=2$ for the smoothed model.
> > >
> > > | Epsilon in                                     |   0.2 |   0.4 |   0.6 |   0.8 |     1 |   1.2 |   1.4 |
> > > |------------------------------------------------|------:|------:|------:|------:|------:|------:|------:|
> > > | Certified cosine similarity                    | 0.919 | 0.784 | 0.629 | 0.522 |  0.46 | 0.383 | 0.341 |
> > > | Smoothing error (cosine distance)              | 0.021 | 0.074 |  0.14 | 0.173 | 0.218 | 0.239 | 0.313 |
> > > | Adversarial cosine similarity (base model)     |     1 | 0.994 | 0.958 |  0.93 | 0.885 | 0.788 | 0.752 |
> > > | Robust adversarial similarity (smoothed model) | 0.995 |  0.99 | 0.986 |  0.98 | 0.974 |  0.97 | 0.966 |
> > >
> > > We observe that as we increase the attack budget, the adversarial cosine similarity for the base model decreases rapidly. The certified cosine similarity also decreases with the increase in the input radius. The empirical robustness of the smoothed model remains high for the range of the input radii we considered, but the error introduced by center smoothing is quite high when the input radius is $\geq 1$. We note that although the certified similarity is lower than the adversarial similarity, the empirical similarity of the smoothed model remains above the adversarial similarity of the base model. It is challenging to generate strong attacks for the metric setting in the range of input radii we considered. However, perturbations with much higher budgets still remain imperceptible for ImageNet images as the dimensionality of the input is very high - 3 x 224 x 224. Below we show the adversarial cosine similarity for the base model for larger epsilon in:
> > >
> > > | Epsilon in                                 |    2 |     3 |     4 |      5 |     10 |
> > > |--------------------------------------------|-----:|------:|------:|-------:|-------:|
> > > | Adversarial cosine similarity (base model) | 0.45 | 0.171 | -0.17 | -0.514 | -0.804 |
> > >
> > > The adversarial attack keeps getting stronger with increasing attack budget, almost flipping the output vector for an $\ell_2$ budget of 10.
> > >
> > > Our method provides good certificates when the input radius is roughly in the range of 0 to 1. This is true for classification tasks as well. The certified accuracy achieved in Certified Adversarial Robustness via Randomized Smoothing by Cohen et.al. for ImageNet classification drops below 20% for an input radius of 2. However, the attacks in the classification setup are much stronger. We would like to note that even though our PGD attack does not make the adversarial similarity for the base model go below the certified similarity for the robust model, there may still exist stronger attacks that can. But the certificates guarantee that no future attack strategies can reduce the cosine similarity for the robust model below the certified similarity. Also, any improvements in randomized smoothing for classification can be leveraged by our method as it does not critically depend on the smoothing distribution and the threat model.
> > >
> > > We would also like to point out that, to the best of our knowledge, previous works on certified defenses for problems in our setting have not shown an attack that degrades the base model’s performance below the certified performance of the robust model in the range of input radii considered, e.g., Scalable Certified Segmentation via Randomized Smoothing by Fischer et. al. and Making medical image reconstruction adversarially robust by Adva Wolf. The certified segmentation work only compares the adversarial and robust segmentation outputs visually and does not use a metric (like IoU) to compare them. Also not that Fischer et. al. produce robust segmentation “outputs” by leaving out some pixels unclassified (white pixels in the certified output). Their certificates are computed w.r.t these partial segmentation outputs. Our center smoothing approach, on the other hand, always produces valid elements in the output space for any problem and the certificate holds for this valid output. Thus, the two certificates are not comparable.
> > >
> > > In a previous version of this paper, we compared center smoothing to a baseline certificate from the article: Making medical image reconstruction adversarially robust by Adva Wolf.
> > > This paper derives a Lipschitz-bound for a smoothed function defined as the element-wise mean of the output vectors when the input is sampled from a Gaussian smoothing distribution. This baseline certificate only holds when the output metric is the $\ell_2$-norm of the difference of the outputs, i.e., $||\bar{f}(x) - \bar{f}(x')||_2$.
> > > Our certificates always outperformed this baseline by a significant margin.
> > > Below we present some of those results ($h = \frac{\epsilon_1}{\sigma} = 2$):
> > >
> > > **Dimensionality Reduction (CIFAR-10):**
> > >
> > > | Epsilon in                 |    0.2 |    0.4 |    0.6 |    0.8 |      1 |
> > > |----------------------------|-------:|-------:|-------:|-------:|-------:|
> > > | Lipschitz bound (baseline) | 37.839 | 37.839 | 37.839 | 37.839 | 37.839 |
> > > | Epsilon out                | 11.438 | 11.546 | 15.385 |  20.02 | 20.569 |
> > > | Smoothing error            |  2.103 |   2.29 |  3.062 |  3.982 |  4.043 |
> > >
> > > **Dimensionality Reduction (MNIST):**
> > >
> > > | Epsilon in                 |     0.2 |     0.4 |     0.6 |     0.8 |       1 |
> > > |----------------------------|--------:|--------:|--------:|--------:|--------:|
> > > | Lipschitz bound (baseline) | 19.1156 | 19.1156 | 19.1156 | 19.1156 | 19.1156 |
> > > | Epsilon out                |   2.958 |   4.743 |   6.537 |   8.384 |   9.896 |
> > > | Smoothing error            |   0.458 |   0.641 |   0.757 |   0.909 |   1.012 |
> > >
> > > **Image Reconstruction (CIFAR-10):**
> > >
> > > | Epsilon in                 |    0.2 |    0.4 |    0.6 |    0.8 |      1 |
> > > |----------------------------|-------:|-------:|-------:|-------:|-------:|
> > > | Lipschitz bound (baseline) | 37.839 | 37.839 | 37.839 | 37.839 | 37.839 |
> > > | Epsilon out                |  6.188 | 11.012 | 12.328 | 16.157 | 18.662 |
> > > | Smoothing error            |  1.037 |  1.938 |  2.128 |  2.678 |  3.015 |
> > >
> > > **Image Reconstruction (MNIST):**
> > >
> > > | Epsilon in                 |     0.2 |     0.4 |     0.6 |     0.8 |       1 |
> > > |----------------------------|--------:|--------:|--------:|--------:|--------:|
> > > | Lipschitz bound (baseline) | 19.1156 | 19.1156 | 19.1156 | 19.1156 | 19.1156 |
> > > | Epsilon out                |   3.182 |   5.622 |   7.387 |   9.682 |  10.459 |
> > > | Smoothing error            |   0.429 |   0.691 |   0.948 |   1.112 |   1.342 |
> > >
> > > Observations: Since $\frac{\epsilon_1}{\sigma}$ is kept constant, the Lipschitzness-based certificate (baseline) also remains constant (refer to theorem 1 of the above article).
> > > The above tables show that the certified output radius produced by center smoothing always remains below the baseline and thus is tighter (lower is better).
> > >
> > > Hope the above results help you make the final decision for this paper. If you have any more concerns, we would really appreciate it if you could let us know as soon as possible.

---

### Official Review · Reviewer_t5kY · 2021-07-14

**Rating:** 6
**Confidence:** 5

**Summary:**

This paper aims to extend certifiable robustness methods based on randomized smoothing into the situation when the output space is a general metric space. The basic idea is to output the center of the smallest ball in the output space that can enclose half the probability mass of the output distribution with Gaussian input.

**Limitations And Societal Impact:**

Not applicable.

**Main Review:**

Originality: The paper is based on randomized smoothing methods but focuses on a new setting where the output space is a general metric space.

Quality: The proposed method is interesting and suitable for many situations. It will be better if the authors can explain why the proposed method will work without requiring the output of the base classifier to be bounded, because most of the previous randomized smoothing approaches usually rely on the assumption that the output is bounded (in [0, 1]).

Clarity: This paper is well written.

Significance: The setting considered in this paper is potential, however, the experiments results in the paper is limited. The reviewer would like to encourage the authors to add more examples (maybe as future work) to showcase the usage of certifying in a metric space.

---
Update after rebuttal:
The rebuttal provides more detailed explanations of the paper which the reviewer suggests should be added to the paper. And I'll keep my score as weak accept.

**Time Spent Reviewing:**

3

---

> ### Author Response · Authors · 2021-08-09
> **Outlier Resistance of Center Smoothing and Additional Applications**
>
> "why the proposed method will work without requiring the output of the base classifier to be bounded ..." -- The output of the base function in our setting are objects like sets, images, graph nodes, etc., for which boundedness is not defined in the same sense as for real-valued outputs. Perhaps we should instead be asking: why do we not require the distance function to be bounded? Center smoothing works by computing the center of the smallest ball that encloses at least half of the probability density of the output distribution. This makes it resistant to outliers, meaning that the smoothed output does not get affected by the distance of a point that is not in the minimum enclosing ball. One could move an outlier point an arbitrarily large distance from the ball without affecting the output of the smoothed function at all. This is the same reason why median smoothing introduced in the paper titled 'Detection as Regression: Certified Object Detection by Median Smoothing' by Chiang et al. is also outlier resistant as long as the outlier remains on the same side of the median. However, if the output is something like the mean of the samples then we would need the output range to be bounded as the mean is affected by the value of every sample.
>
> "The setting considered in this paper is potential, however, the experiments results in the paper is limited. The reviewer would like to encourage the authors to add more examples (maybe as future work) to showcase the usage of certifying in a metric space." --
> In the appendix, we have included additional experiments that certify the output of a GAN that generates high-dimensional ImageNet images under a perceptual distance metric LPIPS. The main challenge here was to make center smoothing work for high-dimensional outputs like ImageNet images. There are other potential directions for future work, e.g., certifying the maximum number of pixels that could change in a segmentation output, certifying models that generate natural language from images/videos like image captioning and scene summarizing, license plate number detection, etc.

---

> > ### Author Response · Authors · 2021-08-31
> > **Additional Example: Perceptual Hash**
> >
> > We would like to follow up on the request for additional examples of certifying in a metric space. Recently, a perceptual hash function NeuralHash used by Apple in their CSAM detection system was attacked using adversarial examples. It was shown that by adding a small perturbation in an image it is possible to drastically change its hash. Perceptual hash functions work by mapping similar looking images to vectors that are close to each other and perceptually different images to vectors that are far apart. In neural network based hashes, these vectors are generated using the high-level features produced by a neural network trained for some task, like classification (say). They are then converted to strings of integers using a technique called Hyperplane Locality Sensitive Hashing. Thus by changing the feature vectors one can alter the final hashes. The similarity between the feature vectors is measured using metrics such as cosine similarity, angular distance, etc. The attacks on NeuralHash sought to find an adversarial example whose feature vectors would have a high dissimilarity (say, by minimizing the cosine similarity) with the feature vector of the clean image.
> >
> > We applied center smoothing to certify these feature vectors under the cosine similarity (1 - cosine distance) metric. A cosine similarity of one means that the two vectors are completely aligned with each other and a similarity of negative one means that they are pointing in opposite directions. We use a pre-trained resnet18 model to generate high-level features (right before softmax) of 224 x 224 ImageNet images. We certify these feature vectors (1000 dimensions) under the cosine distance metric using our method. Note that the cosine distance does not satisfy the triangle inequality, but satisfies an approximate version of it with $\gamma = 2$. However, center smoothing can generate certificates even for such distance measures (see section 4 page 6). Below we report the median certified cosine similarity of the robust model and the median smoothing error (cosine distance between outputs of the base model and the robust model) computed for fifty randomly sampled images from the dataset for different input radii.
> >
> > | Epsilon in                        |   0.2 |   0.4 |   0.6 |   0.8 |     1 |   1.2 |   1.4 |
> > |-----------------------------------|------:|------:|------:|------:|------:|------:|------:|
> > | Certified cosine similarity       | 0.919 | 0.784 | 0.629 | 0.522 |  0.46 | 0.383 | 0.341 |
> > | Smoothing error (cosine distance) | 0.021 | 0.074 |  0.14 | 0.173 | 0.218 | 0.239 | 0.313 |
> >
> > Our method achieves meaningful certificates for the cosine similarity of the robust output under adversarial perturbations of the input image. This further showcases the versatility of our approach. Hope this example helps you in making the final decision on this paper.

---

### Official Review · Reviewer_mjCZ · 2021-07-15

**Rating:** 6
**Confidence:** 4

**Summary:**

This paper proposed a smoothing technique to perform adversarial training, such that the trained model is robust to small $l_2$ perturbations in the input. The major contribution is that the proposed method is applicable to a wide range of distance metrics. The idea of computing the minimum enclosing ball is also a very interesting idea, and I believe similar idea can be used to smoothing based optimization.

**Limitations And Societal Impact:**

Yes.

**Main Review:**

Pros:
1. This paper provides not only a theoretical justification of the proposed method but also an algorithm that can be implemented and used in practice.

Cons:
1. There is no description on the distribution of the input perturbation $x-x'$. It only says $||x-x'||_2 \le \epsilon_1$. Is the noise uniformly distributed into all the pixels or concentrated in a local region. Since the Guassian smoothing is used, it is natural to assume that the proposed method only works when the input image is perturbed by a white noise, unless additional experiments show that the proposed method also works for concentrated perturbations. In recent years, there are new adversarial attack methods, e.g., square attack,  that generate concentrated perturbations to fool an image classifier. Thus, it is important to understand what kind of perturbations the proposed method can handle.

2. In the examples using pre-trained models, they authors trained the base models on data with added noises to make them more robust to input perturbations. My understanding is that this is not part of the proposed smoothing technique. For those examples, it is not clear which part (noise injection or smoothing) plays a more important role in making a robust model. An ablation study on this would be helpful.

3. It's not clear how to choose the value $h$ in $\epsilon_1 = h \sigma$ on Line 241. The authors claim, in Line 246, that the increasing of $h$ improves the quality of the certificates. Does it mean that I should use a very large $h$? In practice, we usually assume we know $\epsilon_1$ and question is how to find an appropriate $\sigma$ so that we can get the best certificates (smallest $\epsilon_2$). I don't know whether this is the setting considered in this paper.

4. There is a little abuse of notation that cause some confusion. For example, is $\sigma$ in Line 237 the same as $\sigma$ in Line 241? Is the constant $h$ in Line 241 the same as the $h$ in Line 96? I think they are all different. If so, it would be better to use different notations.

In summary, even though the authors proposed a promising idea, there is a lack of ablation study to demonstrate the effectiveness of the method. I would be happy to increase my rating if the above concerns are addressed.





**Time Spent Reviewing:**

5

---

> ### Author Response · Authors · 2021-08-10
> **Ablation Study and Other Clarifications**
>
> We would like to begin by clarifying that our method is not an adversarial training procedure as the summary seems to suggest. It is not a procedure to train a robust model. Instead, the center smoothing procedure takes a trained model and produces a smoothed version of it which is provably robust. The base model can be trained using any procedure (including adversarial training). Similar to most previous randomized-smoothing techniques, center smoothing is a model-agnostic procedure.
>
> Below we address the concerns raised in the main review:
>
> 1. "There is no description on the distribution of the input perturbation" -- Our certificates hold for *any* perturbation $x’$ of the original input $x$ as long as $||x - x'||_2 \leq \epsilon_1$ is satisfied as indicated by the $\forall$ quantifier in front of $x'$ in the equation in line 50 (page 2).
> It is common for most existing provable robustness procedures to produce guarantees that hold for all the points in the neighbourhood of the input point $x$, which in our case is the region defined by $N_x$ = { $x'$ s.t. $||x - x'||_2 \leq \epsilon_1$}.
> Thus, the perturbation may be distributed over all pixels or concentrated in a local region, our certificate holds as long as the perturbed input lies in $N_x$.
> In this work, we have restricted the threat model (on input side) to an $\ell_2$-norm bounded adversary and the smoothing distribution to be isometric Gaussian because our focus is on the *output* space.
> We designed a procedure that can certify the output under a wide range of distance metrics.
> However, our method does not critically rely on the $\ell_2$ threat model or the Gaussian smoothing distribution to work.
> This method can be adapted for other threat models and smoothing distribution (e.g., Laplacian noise for $\ell_1$ robustness [1]) to extend the scope of the output space.
>
>     [1] $\ell_1$ adversarial robustness certificates: a randomized smoothing approach, Teng et. al. 2020.
>
> 2. **Ablation study:** Just to clarify, the pre-trained models, MTCNN and BigGAN, that we use for certifying face detection and GAN outputs (included in supplementary material) have not been trained with Gaussian noise (see line 236, page 6). These models were obtained directly from online repositories and were not trained by us.
> We pass these models to our center smoothing procedure along with the desired output metric to obtain certificates.
> The autoencoder and image reconstructor models used for the experiments in sections 5.2 and 5.3 were trained by us with isometric Gaussian noise with different variances.
> This is a common practice in randomized smoothing based techniques to train the base-model with the same kind of noise used for smoothing. It increases the *empirical* robustness of the base-model but it does not yield *provable* robustness just by itself.
> Below we present results from the ablation study for the dimensionality reduction and image reconstruction problems with total variation distance function for $h=2$:
>
> **Dimensionality Reduction (MNIST):**
>
> | Epsilon in                           |    0.2 |     0.4 |     0.6 |     0.8 |       1 |
> |--------------------------------------|-------:|--------:|--------:|--------:|--------:|
> | Epsilon out (w/ training noise)      | 67.495 |   101.5 | 132.483 |   162.3 | 200.486 |
> | Epsilon out (w/o training noise)     |  139.2 | 197.495 |  248.12 | 285.765 | 325.356 |
> | Smoothing error (w/ training noise)  | 10.424 |  13.185 |  15.873 |  18.175 |  24.278 |
> | Smoothing error (w/o training noise) | 35.254 |  47.228 |  54.919 |   64.01 |  74.623 |
>
> **Dimensionality Reduction (CIFAR-10):**
>
> | Epsilon in                           |     0.2 |     0.4 |     0.6 |     0.8 |       1 |
> |--------------------------------------|--------:|--------:|--------:|--------:|--------:|
> | Epsilon out (w/ training noise)      | 554.032 |  597.19 | 696.103 | 842.554 | 821.541 |
> | Epsilon out (w/o training noise)     | 469.004 | 791.567 | 925.972 | 986.154 | 1018.07 |
> | Smoothing error (w/ training noise)  |  95.044 | 120.031 | 136.022 | 166.002 | 152.988 |
> | Smoothing error (w/o training noise) |  96.212 | 173.543 | 234.179 |  278.77 | 296.013 |
>
> **Image Reconstruction (MNIST):**
>
> | Epsilon in                           |    0.2 |     0.4 |     0.6 |     0.8 |       1 |
> |--------------------------------------|-------:|--------:|--------:|--------:|--------:|
> | Epsilon out (w/ training noise)      | 74.389 | 121.833 | 153.172 | 184.343 | 208.222 |
> | Epsilon out (w/o training noise)     | 88.239 | 168.971 | 245.918 | 317.216 | 387.325 |
> | Smoothing error (w/ training noise)  | 11.006 |  17.096 |  20.807 |  23.983 |  26.806 |
> | Smoothing error (w/o training noise) | 13.047 |  27.003 |  39.181 |  51.799 |  62.361 |
>
> **Image Reconstruction (CIFAR-10):**
>
> | Epsilon in                           |     0.2 |     0.4 |     0.6 |      0.8 |       1 |
> |--------------------------------------|--------:|--------:|--------:|---------:|--------:|
> | Epsilon out (w/ training noise)      | 262.983 | 440.158 | 528.451 |  672.049 | 732.281 |
> | Epsilon out (w/o training noise)     | 268.314 | 584.543 |  800.71 | 1111.651 | 1321.97 |
> | Smoothing error (w/ training noise)  |  48.063 |  78.598 |  93.505 |  116.864 | 129.065 |
> | Smoothing error (w/o training noise) |  49.565 | 107.701 |  150.22 |  212.909 |  250.25 |
>
> We observe that the performance of the smoothed models and their certificates deteriorate in the absence of training noise which is as expected.
> However, they still yield meaningful and non-trivial certificates which shows that both center smoothing and training with Gaussian noise contribute towards the robustness and performance of the smoothed models. We will include plots for these results in the updated version.
>
> 3. "It's not clear how to choose the value $h$" -- As our experiments show, increasing $h$ lowers the certified output radius $\epsilon_2$ (for the same value of $\epsilon_1$), thus increasing the quality of the certificate.
> However, as we increase $h$, the smoothing noise $\sigma$ decreases for the same $\epsilon_1$ as $\epsilon_1 = h \sigma$.
> This increases the value of $p$ in equation 4 (line 182, page 5) which in-turn increases $q$ in equation 5 (line 187, page 5).
> But $q$ is a quantile and must be between 0 and 1 (we check for this in the code).
> Thus, increasing $h$ too much will make $q$ greater than 1 causing the procedure to fail. In practice, one would choose the highest $h$ for which the procedure does not fail too often and choose the smoothing noise $\sigma$ accordingly for a given $\epsilon_1$.
> We will include this discussion in the next version of the paper.
>
> 4. "There is a little abuse of notation that cause some confusion." -- Indeed, there has been collision of some notations and we apologize for the confusion. As correctly pointed out, the $\sigma$ in line 237 (training noise) is different from the $\sigma$ in line 241 (smoothing noise) and the $h$ in line 241 (the ratio between $\epsilon_1$ and $\sigma$) is different from the one in line 96 (which denotes a function).
> Thank you for pointing them out. We will make sure to correct them in the next update.

---

> > ### Author Response · Authors · 2021-08-26
> > **Looking forward to your feedback**
> >
> > Dear Reviewer mjCZ:
> >
> > It has been a while since we submitted our response to your review and we were wondering if you have had a chance to go through it. We have added the ablation study as you requested and provided clarifications for your questions. We would like to know if we have addressed all of your concerns satisfactorily. If there is anything more we could do to help you make your final decision about this paper, please let us know.
> >
> > Thank you!

---

> > ### Comment · Reviewer_mjCZ · 2021-08-30
> > **Thank you**
> >
> > I appreciate the authors' feedback. It addresses most of my concerns, so I will increase my score.

---

> > > ### Author Response · Authors · 2021-08-30
> > > **Thank you for the positive feedback!**
> > >
> > > Thank you for your comments and score increase! We are glad our response addressed your concerns.

---

### Decision · Program_Chairs · 2021-09-27

**Decision:**

Accept (Poster)

**Comment:**

Thank you for your submission to NeurIPS.  The reviewers and I are all in agreement that the proposed work presents a nice (if incremental, at least from a technical perspective), extension to randomize smoothing, to apply to more general output spaces than previous approaches.  The results are technically sound, even if by no means represent a foundational breakthrough, and the experiments illustrate the potential usefulness of the approach.  And the authors did a great job during the response period discussing the issues raised by the reviewers and offering further potential benefits of the approach.

The majority of the discussion centered around the question of whether such a result is substantial enough (from a philosophical standpoint) to warrant publication at NeurIPS.  And in this setting, I want to make two points.  First, I think it's entirely appropriate for the reviewer to bring up the point in this manner.  Paper selection for NeurIPS _is_ ultimately about making subjective (rather than just factual) evaluations of papers, and it's entirely appropriate for the reviewer to highlight their concerns about the fundamental novelty and perceived impact of the work.  Indeed, it's far preferable to bring it up in an honest and frank manner (and engage fully with the reviewers afterward in discussion), than to try to couch this fundamental concern in the veil of minor technical issues with the paper (which is too often the standard in reviews).  If there are questions about a paper's significance, it can and should be brought up in the manner.  And second, I must say that I agree to some extent with what this reviewer is saying: there _is_ an extent to which the paper seems to be extending randomized smoothing to "one more domain".  And the other three reviewers, while being positive, are only minorly so, and weren't substantially swayed by the author response.

Despite this, however, I am going to recommend the paper be accepted.  While I'm sympathetic to the points that the negative reviewer makes, ultimately I think that rejecting an otherwise-universally-recommended-to-be-accepted paper on the grounds of perceived significance be an incorrect decision, given that it's such a subjective quality.  My own reading of the paper is that it presents a practically-valuable extension to randomized smoothing for problems beyond classification, using technically sound approaches.  Although the methodology may not be the most fundamentally groundbreaking, it covers a topic of likely interest to a wide variety of practitioners who may be considering using randomized smoothing, and hence has a notable value for the field.  Given all this, I believe the paper does meet the bar for NeurIPS despite the ultimate persisting disagreement between reviewers.